# A conservative immersed boundary method for the multi-physics urban large-eddy simulation model uDALES v2.0

**Sam O. Owens**[1], **Dipanjan Majumdar**[1], **Chris E. Wilson**[1], **Paul Bartholomew**[2], and **Maarten van Reeuwijk**[1]

[1]Department of Civil and Environmental Engineering, Imperial College London, London SW7 2AZ, United Kingdom
[2]EPCC, University of Edinburgh, Edinburgh EH8 9BT, United Kingdom

**Correspondence:** Sam O. Owens (sam.owens18@imperial.ac.uk)

**Abstract.** uDALES is an open-source multi-physics microscale urban modelling tool, capable of performing large-eddy simulation (LES) of urban airflow, heat transfer, and pollutant dispersion. We present uDALES v2.0, which has two main new features: (1) an improved parallelisation that prepares the codebase for conducting exascale simulations and (2) a conservative immersed boundary method (IBM) suitable for an urban surface that does not need to be aligned with the underlying Cartesian grid. The urban geometry and local topography are incorporated via a triangulated surface with a resolution that is independent of the fluid grid. The IBM developed here includes the use of wall functions to apply surface fluxes, and the exchange of heat and moisture between the surface and the air is conservative by construction. We perform a number of validation simulations, ranging from neutral, coupled internal–external flows and non-neutral cases. We observe close agreement with the relevant literature, both in cases in which the buildings are aligned with the Cartesian grid and when they are at an angle. We introduce a validation case specifically for urban applications, for which we show that supporting non-grid-aligned geometries is crucial when solving surface energy balances, with errors of up to 20 % associated with using a grid-aligned geometry.

## 1 Introduction

It is expected that the global average temperature will increase by at least 1.5 °C by 2050 (Masson-Delmotte et al., 2022) and that the fraction of people living in cities will increase from around 50 % today to 58 % over the next 50 years (United Nations Human Settlements Programme (UN-Habitat), 2022). To illustrate, by 2050 the climate of London, UK, is expected to resemble that of Barcelona, Spain (Bastin et al., 2019), and could be home to over 1 million more people (Greater London Authority, 2023). In this context, urban residents face particular environmental challenges, e.g. the urban heat island effect, in which the temperature in cities tends to be higher than their surrounding rural area (Oke et al., 2017). Excess heat, particularly when coupled with humidity, is uncomfortable at best and lethal at worst, and it can damage infrastructure and impact wildlife. Moreover, tall buildings can exhibit areas of high wind speed at their base, which can be a safety concern (Xu et al., 2017). Cities also tend to have poorer air quality, largely due to greater emission of pollutants within the city itself (Enenkel et al., 2020). Given the trends with respect to climate change and urbanisation, these issues are likely to become ever more salient. It is therefore crucial to study airflow, heat transfer, and pollutant dispersion in the built environment in order to provide insight into urban microclimate and air quality.

To study these processes, a combination of observations, experiments (physical modelling), and numerical modelling is used. Given the increasing capabilities of high-performance computing (HPC), numerical models are being developed to take advantage of these resources. These models operate at a range of scales, from the mesoscale ($\mathcal{O}(1\,\text{km})$), at which the effect of urban areas are parameterised (Walters et al., 2019; Skamarock et al., 2019), to the microscale ($\mathcal{O}(1\,\text{m})$), at which the surface is resolved at high resolution. Airflow (and transport of heat, moisture, and pollutants) is commonly modelled using computational fluid dynamics (CFD). Pollutant dispersion also potentially involves

modelling the chemical reactions of various aerosols. Incorporating heat transfer at the surface requires modelling the surface energy balance, i.e. the net radiation, convection, evaporation, and conduction at the surface–atmosphere interface.

There are standalone tools for modelling airflow, heat transfer, and pollutant dispersion individually, but as these processes fundamentally interact, multi-physics modelling tools have been developed that have some degree of coupling between them. Some of these tools are particularly focused on microclimate and pedestrian comfort in realistic urban environments. These tend to have advanced thermal models, typically employing ray tracing for radiation. Examples include ENVI-met (Huttner, 2012), MITRAS (Salim et al., 2018), SOLENE-Microclimate (Musy et al., 2015, 2021), urbanMicroclimateFoam (Kubilay et al., 2018), and the model described in Wong et al. (2021). However, these models solve the Reynolds-averaged Navier–Stokes (RANS) equations and typically parameterise all turbulence using a one-point closure (Pope, 2001). It is challenging for RANS models to accurately simulate stratified or convective atmospheric conditions (Hanjalić, 2002). Moreover, these models often fail to reproduce unsteady flow features, leading to inaccurate prediction of quantities like turbulent kinetic energy for flows that involve time-dependent modulation (Moonen et al., 2012; van Hooff et al., 2017; Blocken, 2018; Vita et al., 2020). Two-moment closures and unsteady RANS approaches have been developed to address these problems, but these are rarely used in urban applications (Toparlar et al., 2015; Tominaga, 2015; Gao et al., 2018; Antoniou et al., 2019; Hadžiabdić et al., 2022). Conversely, large-eddy simulation (LES) resolves the mean flow as well as all turbulence with a length scale greater than the filter size (often equal to the grid resolution), with the small-scale turbulence being modelled using subgrid-scale models. Since LES resolves the energetic scales of the turbulence, it is expected to be more accurate than RANS (Murakami, 1993; Baker, 2007), albeit with a significant increase in computational cost (Chen, 2009). This is provided that the boundary conditions, including the surface, are modelled accurately, which is challenging for urban boundary layer flows (Radovic et al., 2023).

With both RANS and LES, it is possible to use an unstructured mesh (fitted to the geometry). However, with LES another popular approach is to use a Cartesian grid combined with an immersed boundary method (IBM; Verzicco, 2023), which avoids challenges associated with unstructured meshes such as the effect of cell geometry (Hefny and Ooka, 2009). More importantly, using a Cartesian grid increases computational performance substantially because solving for the pressure can be carried out using the fast Fourier transform (FFT) algorithm (Ferziger et al., 2020). Traditionally the IBM is discussed in the context of direct numerical simulation (DNS) approaches; LES has different requirements because unlike DNS, the flow near the obstacle is not fully resolved (Verzicco, 2023). Urban LES models that use a Cartesian grid with an IBM have typically evolved from atmospheric flow solvers, and some are equipped with multiphysics surface schemes capturing urban processes, for example, PALM-USM (Resler et al., 2017; Maronga et al., 2020; Krč et al., 2021), City-LES (Watanabe et al., 2021), and uDALES v1.0 (Suter et al., 2022). These models can currently only specify walls that align with the Cartesian grid, so urban geometries that do not actually meet this requirement are approximated as doing so using a voxel representation. To illustrate, for a case used to demonstrate the capabilities of uDALES v1.0 (presented in Sect. 4.3 of Suter et al., 2022), the geometry was generated using a raster of a surface elevation map (obtained from GIS data), and the shape of buildings that are not aligned with the grid is distorted. This is problematic for realistic, complex geometries, particularly with respect to radiation as this is strongly influenced by the surface area and orientation (Salim et al., 2022). The net shortwave radiation on the buildings for this case is shown in Fig. 1; note the non-physical pattern visible on the walls that have been distorted by the voxelisation.

The aim of this paper is to present a new version of uDALES, as v1.0 is limited in several key ways. As previously mentioned, the IBM is only capable of modelling walls that are aligned with the Cartesian computational grid, which hinders its ability to faithfully represent realistic urban geometries. Also, the code is parallelised with a one-dimensional domain decomposition implemented with bespoke routines using the message passing interface (MPI) library, which means it cannot effectively utilise the large number of processors available on HPC systems. The final limitation is that it is not possible to use the FFT algorithm to solve for the pressure with inflow–outflow boundary conditions, though it is possible to use the FFT in periodic situations.

uDALES v2.0 has been primarily developed to address these key issues. The geometry is specified as an unstructured triangulated surface that is given independently of the grid using the STL file format, and a novel technique for applying surface momentum and heat fluxes given this non-grid-aligned approach has been implemented for the IBM. The code is parallelised with a two-dimensional domain decomposition, implemented using the 2DECOMP&FFT library (Li and Laizet, 2010; Rolfo et al., 2023), thus enabling it to run on many more processors than uDALES v1.0. Finally, it is capable of using an FFT-based method to solve for the pressure when using inflow–outflow boundary conditions, thereby enhancing the performance of simulations employing these conditions. In addition to these major changes, uDALES v2.0 also has an improved algorithm for calculating shortwave radiation, as well as a novel method to prevent the formation of artificial flow features in periodic simulations. A summary of the upgrades is given in Table 1.

The paper is structured as follows. Section 2 gives an overview of uDALES, Sect. 3 describes the upgrades that together make up uDALES v2.0, and Sect. 4 presents an

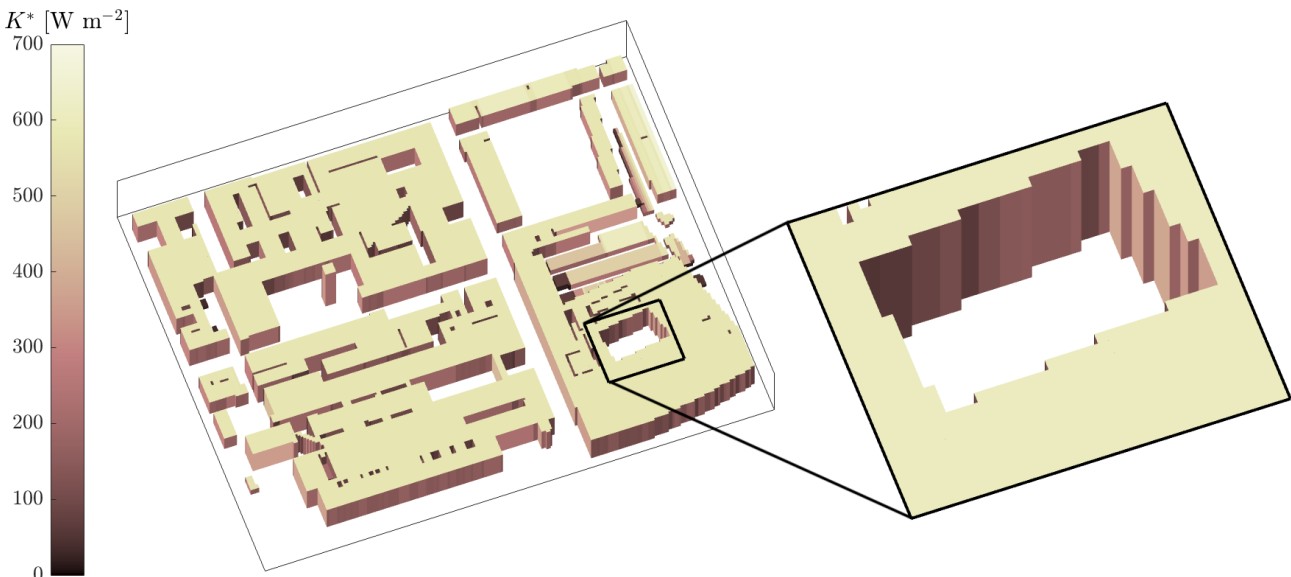

**Figure 1.** Net shortwave radiation ($K^*$) on buildings for the demonstration case in Suter et al. (2022). Note that the ground surfaces have been excluded from the visualisation for clarity.

**Table 1.** Comparison of upgraded features between uDALES v1.0 and v2.0.

| Feature | uDALES v1.0 | uDALES v2.0 |
|---|---|---|
| Domain decomposition | One-dimensional | Two-dimensional |
| Pressure solver | Only entirely FFT-based for periodic cases | Entirely FFT-based for periodic and inflow–outflow cases |
| Geometry representation/IBM | Blocks (voxels) – grid-aligned | Triangulated surface – grid-independent |
| Shortwave radiation algorithm | Inaccurate shading | Accurate shading |
| Periodic (precursor) simulations | No method | Method to avoid permanent streamwise super-structures |

evaluation of the model by comparing it with uDALES v1.0 and other physical and numerical models in canonical urban cases. Finally in Sect. 5, applications, limitations, and future development of the model are discussed.

## 2 Model overview

uDALES is an LES model designed to simulate airflow, heat transfer, and pollutant dispersion in urban environments. The governing equations for the LES are unchanged from Suter et al. (2022) and are discretised using the finite-difference method on a Cartesian grid. The presence of obstacles is modelled using the IBM approach, and wall functions are used to parameterise processes near the immersed boundary. Solid surfaces are divided into facets, each with user-defined material properties. The surface energy balance of each facet can be modelled in a two-way-coupled manner with the flow, enabling facet temperature and moisture content (if vegetative) to be evolved. There is also the capability to model trees in terms of drag, deposition of pollutants, and heat transfer. uDALES has been used for studies concerning atmospheric boundary layer processes (Grylls et al., 2020),

air quality (Grylls et al., 2019; Lim et al., 2022), the effect of trees (Grylls and van Reeuwijk, 2022), and parameterisations for larger-scale models (Sützl et al., 2021a, b). The model has been described in detail in Suter et al. (2022), so only novel or improved aspects and relevant concepts are discussed here.

### 2.1 Large-eddy simulation

uDALES solves the (filtered) incompressible Navier–Stokes equations in the Boussinesq approximation:

$$\frac{\partial u_i}{\partial t} = -\frac{\partial p}{\partial x_i} - \frac{\partial u_j u_i}{\partial x_j} + \frac{\partial}{\partial x_j}\left(K_{\mathrm{m}}\left(\frac{\partial u_i}{\partial x_j} + \frac{\partial u_j}{\partial x_i}\right)\right)$$
$$+ \frac{\theta_{\mathrm{v}} - \langle\theta_{\mathrm{v}}\rangle}{\langle\theta_{\mathrm{v}}\rangle}g\delta_{i3} + \mathcal{F}_i + S_{u_i}, \tag{1}$$

where $u_i$ is the filtered (Cartesian) velocity component in the $x_i$ direction, $p$ is the deviatoric kinematic pressure, $\mathcal{F}_i$ is a large-scale forcing, $S_{u_i}$ represents local sources and sinks (e.g. due to wall shear stress), $\theta_{\mathrm{v}}$ is the virtual potential temperature, $K_{\mathrm{m}}$ is the turbulent eddy viscosity calculated using the Vreman (2004) subgrid model, and $\langle\cdot\rangle$ denotes the $xy$-plane average. The pressure is solved using a two-step correction method (Ferziger et al., 2020). Scalar quantities $\varphi$,

such as the liquid potential temperature ($\theta_l$), total specific humidity ($q_t$), and pollutant concentrations ($c$), evolve according to

$$\frac{\partial \varphi}{\partial t} = -\frac{\partial u_i \varphi}{\partial x_j} + \frac{\partial}{\partial x_j}\left(K_h \frac{\partial \varphi}{\partial x_j}\right) + S_\varphi, \qquad (2)$$

where $K_h = K_m/Pr_t$ is the turbulent eddy diffusivity, $Pr_t$ is the turbulent Prandtl number (a model parameter), and $S_\varphi$ represents local scalar sinks and sources (e.g. due to surface heat flux or chemical reactions).

The equations are discretised in space on an Arakawa-C grid (Arakawa and Lamb, 1977), which entails defining pressure and scalars at cell centres and defining the three velocity components ($u_x, u_y, u_z$) at the lower cell face in each direction ($x, y, z$) in a staggered manner. Second-order central difference schemes are used for all prognostic fields except pollutant concentrations, for which a $\kappa$-advection scheme (Hundsdorfer et al., 1995) is employed to ensure monotonicity. Time integration is carried out by a third-order Runge–Kutta scheme. The $y$ grid is always equidistant, whilst the $z$ grid may be stretched. In uDALES v1.0, it is also possible to have a stretched grid in the $x$ direction, though this is rarely used in practice due to considerations with the pressure solver that are discussed in Sect. 3.2. The IBM is used to set appropriate boundary conditions for solid regions, and wall functions provide surface shear stresses and sensible and latent heat fluxes for fluid cells with solid neighbours.

When using inflow–outflow boundary conditions, the inlet can be defined using either a fixed profile (for laminar inflows) or a time-varying plane generated by a precursor simulation, which is only possible for the $x$ direction. The resolution in the $y$ and $z$ directions of the main simulation necessarily equals that of the precursor, meaning no interpolation is required in these directions, and the flow variables on the output plane of the precursor can be copied directly to the inlet of the main simulation. Technically, the Dirichlet inflow boundary condition for the main simulation is enforced at $x = 0$, so $u$ can be set directly as it is defined on cell edges. The other variables are (linearly) interpolated in the $x$ direction as they are defined on cell centres. The outflow condition is convective, meaning the variables are advected by the vertically averaged velocity at the outlet. It is good practice to have a reasonable distance between the boundaries and the object of interest in order to allow the flow to adjust (Tominaga et al., 2008). In this text, $L_x$, $L_y$, and $L_z$ denote the domain length in the $x$, $y$, and $z$ directions, respectively.

## 2.2 Wall functions

Urban LES requires wall functions to model surface fluxes since the typical resolution is insufficient to resolve the wall boundary layers. Wall functions are parameterisations that use velocity and scalar values at the cells closest to the wall, as well as the conditions at the wall, to predict the wall shear stress $\tau_w$, sensible heat flux $H$, and latent heat flux $E$ (Louis, 1979; Uno et al., 1995; Jarvis, 1976; Stewart, 1988). The exact form of the wall functions used in uDALES is given in Suter et al. (2022); Suter (2018). Here, a conceptual description is provided, which is illustrated in Fig. 2. For a given fluid cell adjacent to the boundary, the streamwise velocity, liquid potential temperature, surface temperature, and distance to the wall are denoted as $u_a$, $\theta_a$, $T_s$, and $d$, respectively. The flux of momentum is given by

$$\tau_w/\rho \equiv u_\tau^2 = f_m(d, u_a, \theta_a, T_s; z_0, z_{0,h}), \qquad (3)$$

where $f_m$ is the wall function for momentum and $\rho$ is a reference air density. The momentum roughness length ($z_0$) and heat roughness length ($z_{0,h}$) are parameters specified for each facet. Similarly, the flux of potential temperature is given by

$$H/\rho c_p \equiv u_\tau \theta_\tau = f_h(d, u_a, \theta_a, T_s; z_0, z_{0,h}, Pr_t), \qquad (4)$$

where $f_h$ is the wall function for temperature and $c_p$ is the air specific heat capacity. Finally, denoting the atmospheric specific humidity, surface saturation humidity (itself dependent on $T_s$), and surface relative humidity by $q_a$, $q_{sat}$, and RH, respectively, the corresponding specific humidity flux is given by

$$E/\rho L_v \equiv u_\tau q_\tau = f_e(u_a, q_a, T_s, q_{sat}, \text{RH}, C_h; r_{can}, r_{soil}), \quad (5)$$

where $f_e$ is the wall function for specific humidity, $L_v$ is the air latent heat of vaporisation, $C_h \equiv u_\tau \theta_\tau / u_a (T_s - \theta_a)$ is the heat transfer coefficient, and $r_{can}$ and $r_{soil}$ are the canopy and soil resistances, respectively, which are model parameters.

The fluxes of momentum, heat, and humidity are converted to volumetric source or sink terms ($S_{u_i}$, $S_{\theta_l}$, and $S_{q_t}$, respectively) by multiplying by the surface area and dividing by the volume over which they act, with the relevant area being the fraction of the adjacent facet that is within the fluid cell of interest, and the volume is that of the cell itself. For example, in Fig. 2, the fluid cell depicted with a dotted line experiences fluxes from facet 3. In this 2D representation, half of facet 3 is inside the cell, so this is the area used in determining the source and sink terms. Note that this must be considered separately for each of the staggered grids.

## 2.3 Surface energy balance scheme

The surface energy balance is written in terms of the heat fluxes (in $\text{Wm}^{-2}$) at the surface of each facet:

$$K^* + L^* = H + E + G_0. \qquad (6)$$

The net shortwave radiation ($K^*$) and net longwave radiation ($L^*$) radiation are calculated using the radiosity approach, which assumes all reflection is diffuse and thereby enables the use of view factors (Oke et al., 2017). The albedo ($\alpha$) and emissivity ($\epsilon$) are accordingly specified for each facet. The sensible and latent heat fluxes ($H$ and $E$, respectively) are the average of the fluxes at neighbouring

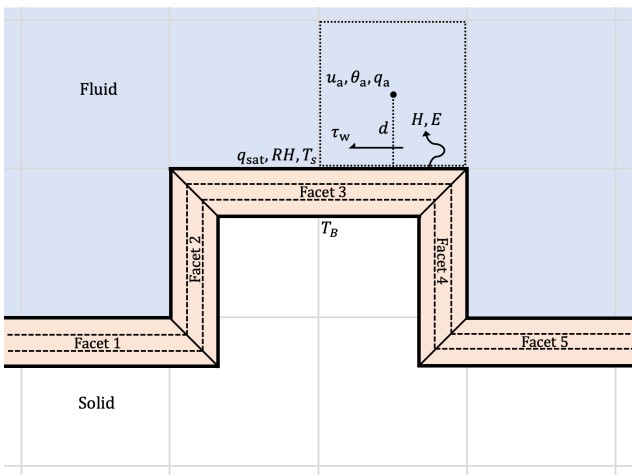

**Figure 2.** Two-way-coupled wall functions and surface energy balance in uDALES. Fluid cells adjacent to the boundary experience a wall shear stress ($\tau_\mathrm{w}$) and fluxes of sensible and latent heat ($H$ and $E$), which are determined by wall functions. These fluxes act as volumetric sources or sinks in the fluid governing equations and simultaneously contribute to the surface energy balance for the corresponding facet. Note that facets are not in fact trapezoidal – this representation is used to avoid them overlapping in this diagram.

fluid cells calculated by the wall functions (see Sect. 2.2), with opposite sign to that experienced by the fluid. The surface conduction ($G_0$) is defined by $G_0 = -\lambda \frac{\partial T}{\partial \xi}|_{\xi=0}$, where $T(\xi)$ is the facet temperature, $\xi$ is the depth into the facet, and $\lambda(\xi)$ is the thermal conductivity. Equation (6) acts as the exterior boundary condition for a 1D enthalpy equation:

$$C\frac{\partial T}{\partial t} = \frac{\partial}{\partial \xi}\left(\lambda \frac{\partial T}{\partial \xi}\right), \tag{7}$$

where $C(\xi)$ is the volumetric heat capacity. Equation (7) is solved numerically for $T$ by discretising each facet into a number of layers along the depth (shown using dashed lines in Fig. 2), with an isothermal boundary condition at the interior (denoted $T_\mathrm{B}$ in Fig. 2). The solution is assumed to be a piecewise continuous quadratic polynomial; see Suter et al. (2022) for further detail. The properties of each layer ($C$, $\lambda$ and thickness) can be specified freely for each facet.

Note that the turbulent fluxes ($H$ and $E$) are calculated at every LES time step, but the surface energy balance is generally evolved using a larger time step than the LES. The fluxes used to solve for the facet temperatures are therefore the time average of the fluxes at LES time steps.

# 3 Upgraded features

## 3.1 Two-dimensional domain decomposition

uDALES v1.0 has a one-dimensional domain decomposition, which involves splitting the computational domain into "slabs" along a single direction – the $y$ direction in this case. Each processor (MPI rank) generally stores and manipulates data structures associated only within a single slab. This means the maximum number of processors that can be used is limited to the number of grid cells in the $y$ direction ($N_y$). Given that typical use cases have $N_y$ equal to 256–1024 and a single node on ARCHER2 (the UK National Supercomputing Service) has 128 processors available, simulations with uDALES v1.0 may only use 10 nodes at most of the thousands that are available (EPCC, 2022).

A two-dimensional decomposition means the domain is divided into "pencils". This is achieved in uDALES v2.0 using the 2DECOMP&FFT library (Li and Laizet, 2010; Rolfo et al., 2023), which is also used by the Xcompact3d framework (Bartholomew et al., 2020). The default pencil orientation is in the $z$ direction, meaning the $x$ and $y$ directions are parallelised as shown in Fig. 3. The extra dimension of parallelisation compared to uDALES v1.0 means that it is possible to run simulations with many more processors, which practically increases the scaling capabilities of the code. The maximum number of processors that it is possible to use is equal to $N_x \times N_y$, where $N_x$ is the number of cells in the $x$ direction. In addition to the computational cells within the pencil, each rank also stores the cells just outside it. These are called halo cells and are stored because they are part of the finite-difference stencil for cells at the edge of the pencil. This naturally requires communication between adjacent ranks every time step. The only time when data are transposed from the $z$ pencil to $x$ and $y$ pencils is while solving for the pressure, because the full extent of the data in a particular direction is required for performing the Fourier transform for that direction.

Figure 4 shows the strong scaling for cases of various sizes and boundary conditions, which were run on ARCHER2 using 128 processors per node. To test the lower-level routines without any novel features, cases without any geometry using periodic boundary conditions were run using grid sizes of $N_x = N_y = N_z = 1024$ and $N_x = N_y = N_z = 2048$. The domain size is $L_x = L_y = L_z = 2048\,\mathrm{m}$, meaning these problem sizes are representative of use cases that involve modelling a convective atmospheric boundary layer at $\mathcal{O}(1\,\mathrm{m})$ resolution. The speed-up for each case is shown in Fig. 4a and b, respectively. The smaller case requires at least 8 nodes (in terms of memory usage), and a good parallel efficiency is observed up to 128 nodes. The larger case requires at least 64 nodes and achieves reasonable efficiency, though less so than the smaller case. Note that all but one (the $1024^3$ case run with 1024 processors) are impossible to run using

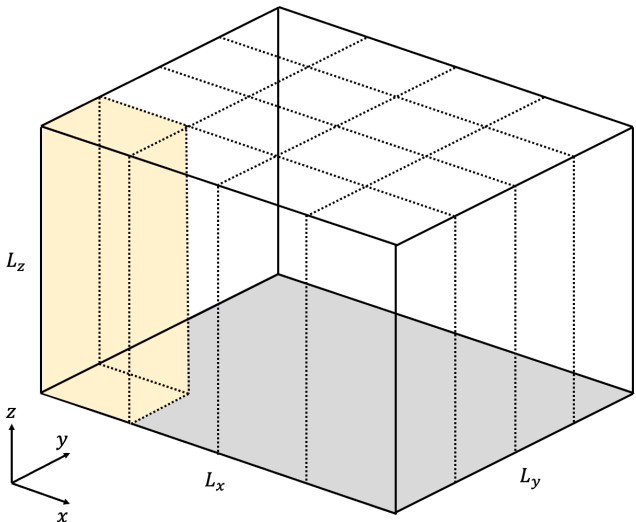

**Figure 3.** A two-dimensional domain decomposition. In this case, the $x$ and $y$ directions are decomposed across 4 processors, so there are 16 in total. The pencil operated on by a single processor is highlighted in yellow.

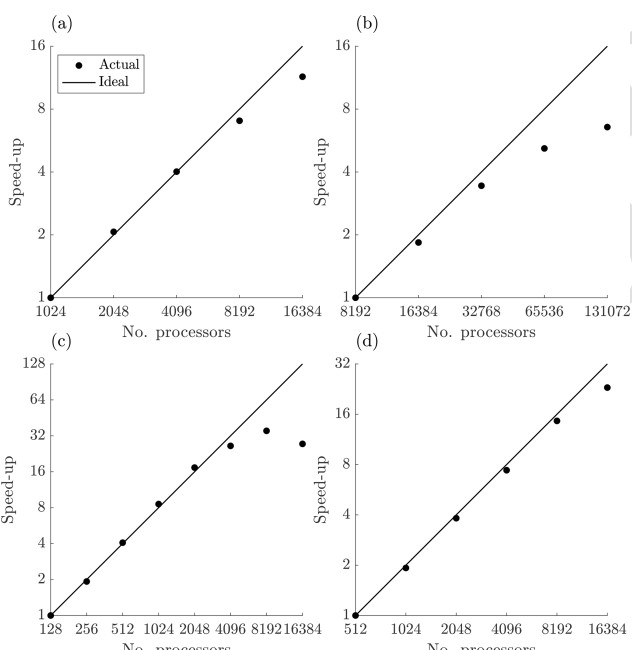

**Figure 4.** Strong scaling for cases: without IBM and periodic boundary conditions using **(a)** $1024^3$ cells and **(b)** $2048^3$ cells; with IBM and inflow–outflow boundary conditions using **(c)** $640 \times 384 \times 384$ cells and **(d)** $1280 \times 768 \times 768$ cells.

uDALES v1.0 due to the number of processors exceeding the number of cells in a single direction.

To test the key new features of uDALES v2.0, i.e. the IBM and fully FFT-based pressure solver for inflow–outflow boundary conditions, cases similar to the validation case shown in Sect. 4.3 (the original, non-rotated case) were simulated. For scaling purposes the resolution is higher, but temperature is not solved for, and the domain width is smaller (the domain size is $L_x \times L_y \times L_z = 190\,\text{m} \times 114\,\text{m} \times 114\,\text{m}$). The smaller case has a grid size of $N_x \times N_y \times N_z = 640 \times 384 \times 384$, and the larger case has a grid size of $N_x \times N_y \times N_z = 1280 \times 768 \times 768$. The speed-up for each case is shown in Fig. 4c and d, respectively. It is possible to run the smaller case on a single node, and good parallel efficiency is observed until 8192 processors are reached, beyond which extra parallelisation in fact becomes detrimental. The larger case required 4 nodes minimum and demonstrated good scaling up to 16 384 cores. These results demonstrate that the limit of parallelisation of uDALES has been increased substantially, including for simulations using the upgraded features of the model. Moreover, 2DECOMP&FFT has been developed for cross-platform usage and has been tested on all major supercomputer architectures (Li and Laizet, 2010; Rolfo et al., 2023), ensuring excellent portability of the code.

### 3.2 Pressure solver

The pressure correction is defined as the difference in pressure between successive time steps, and this quantity satisfies a Poisson equation, which is derived in Appendix A. The Poisson equation in uDALES is solved most efficiently using the FFT in the $x$ and $y$ directions and Gaussian elimination (GE) in the $z$ direction. The reason for not using the FFT in the $z$ direction as well is because GE supports a non-equidistant grid (unlike the FFT), and there is not likely to be a performance gain using the FFT rather than Gaussian elimination due to their algorithmic complexity ($\mathcal{O}(N \log N)$ versus $\mathcal{O}(N)$, respectively). The velocity boundary conditions in the $x$ and $y$ directions determine the specific type of transform performed. If velocity is periodic then the pressure is also periodic and the regular discrete Fourier transform (DFT) is used. If the velocity is inflow–outflow then the pressure has a Neumann boundary condition and a discrete cosine transform (DCT; for a staggered grid) is used. The details of these transforms can be found in Schumann and Sweet (1988); uDALES v2.0 uses the FFTW library to implement them (Frigo and Johnson, 1998). To solve for each direction requires transposing data between the three pencil orientations, which is achieved using routines from the 2DECOMP&FFT library (Li and Laizet, 2010; Rolfo et al., 2023).

Note that uDALES v1.0 does not use the DCT for inflow–outflow situations; instead the cyclic reduction (CR) algorithm is used to solve the $x$ and $z$ directions, while the FFT is used in the $y$ direction (meaning the $y$ direction is always periodic). A non-equidistant grid can be used with CR, hence the ability to use a stretched $x$ grid in uDALES v1.0. However, the CR-based solver is slower than the fully FFT-based solver and is only practical to use with a 1D domain decomposition because CR requires the full extent of the data in both the $x$ and $z$ directions. This is why the inflow–outflow

cases are solved using the DCT in uDALES v2.0. When using inflow–outflow lateral boundary conditions, the boundary condition at the top of the domain must be treated with care. In periodic simulations, there is a no-penetration condition, but with inflow–outflow lateral boundaries, fluid must be allowed to pass through, which is equivalent to setting a non-zero vertical velocity ($u_z$). The physical reasoning for this is that the incoming flow responds to the geometry, which may result in net motion in the $z$ direction. This non-zero $u_z$ is determined by the plane-averaged pressure gradient, and without it, the flow is not incompressible at the top, which is unacceptable both numerically and physically. Further detail on solving the Poisson equation and applying the top boundary condition can be found in Appendix A.

### 3.3   Geometry representation

The geometry in uDALES v1.0 is specified as a set of cuboid blocks with a given size in each Cartesian direction, and the faces of the blocks correspond to the facets. This can be classified as a voxel representation, with each block effectively composed of a set of voxels. The approach is fundamentally limited as it requires the facets to align with the edges of computational cells, which is problematic when the geometry of interest does not in fact do so, as is frequently the case for realistic urban environments (see Fig. 1). Also, the block format is bespoke, which makes it impossible to import geometries directly from other software packages. Instead, custom tools must be relied upon to perform the necessary voxelisation (from a 3D model) or rasterisation and extrusion (from a 2D surface elevation map).

The geometry in uDALES v2.0 is specified as a triangulated surface using an STL file, which consists of a list of triangles in terms of their three vertices in 3D Cartesian space, as well as their surface normals. This formatting is clearly independent of any grid specification and is widely supported by other software packages. By treating each triangle as a facet in uDALES, the model can use this representation to capture non-grid-aligned geometries more faithfully in comparison to a voxel representation and to leverage the tools available to manipulate them. Note that trees are not specified using facets; therefore, the descriptions in this section do not apply to them.

For simple, grid-aligned cases, it may require more triangles to represent the geometry than it would using rectangles, which potentially impacts performance. However, the performance of the IBM does not directly scale with the number of facets, and the performance of the algorithms that do has been upgraded, namely those used in the calculation of shortwave radiation (see Sect. 3.4).

### 3.3.1   Conservative immersed boundary method

Immersed boundary methods (IBMs) are used to model flow around complex geometries while benefitting from the advantages of using a Cartesian grid (Verzicco, 2023). In general, IBMs modify the governing equations of cells inside and/or close to the immersed boundary in order to set the desired flow conditions. uDALES assumes the geometry is non-porous and stationary, which is modelled using no-slip and no-penetration boundary conditions. The fluxes of momentum and scalars across boundaries are prescribed by wall functions, rather than being determined by the advection and diffusion terms in the governing equations (Eqs. 1 and 2). This condition, i.e. zero flux other than what is prescribed by the wall functions, is particularly important for scalars; the IBM must be conservative, meaning there is no net transfer of heat, moisture, or pollutant concentration between the fluid and solid regions due to advection and diffusion processes.

In uDALES v2.0, each grid cell (for each of the staggered grids) is categorised as either fluid or solid depending on whether the cell centre is, respectively, outside or inside the triangulated surface that defines the geometry. This has been achieved using the ray-casting method (Borazjani et al., 2008). Note that in this text, "point" is often used to refer to a cell centre. In non-grid-aligned situations, a tolerance is used, meaning points that are in the fluid region but very close to the surface are classified as solid. This is in part for numerical reasons that are discussed in Sect. 3.3.2 and Appendix B. In addition to this, urban surfaces are aerodynamically rough, meaning it is physically reasonable to classify as solid the points for which the distance to the surface is approximately equal to the size of the roughness elements. Boundary points for both types are identified as those with at least one neighbour of the other type.

At the solid points on each velocity grid, the corresponding velocity component is set to zero. At the fluid boundary points on the velocity and scalar grids, the corresponding governing equations are modified such that any flux contribution due to advection and diffusion from neighbouring solid points is negated. This means that when the facets are aligned with the cell edges, the desired boundary conditions are satisfied exactly. If not, they must be interpreted as follows; the no-slip and no-penetration conditions mean the velocity at solid points is zero, and the zero flux condition (due to advection and diffusion) applies locally between each pair of fluid–solid neighbours. Therefore, according to these conditions, the boundary is located literally at the interface of the fluid and solid cells, aligned with the cell edges, as shown in Fig. 5. This aspect of the IBM is conceptually unchanged from uDALES v1.0, though was implemented from scratch in uDALES v2.0 in order to accommodate the 2D domain decomposition and the triangulated surface geometry representation.

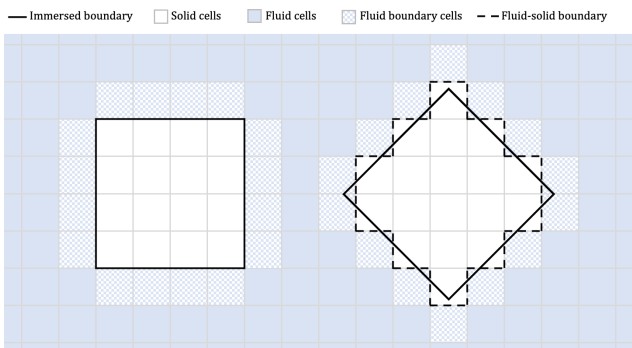

**Figure 5.** A 2D geometry depicted for grid-aligned (left) and non-grid-aligned (right) situations.

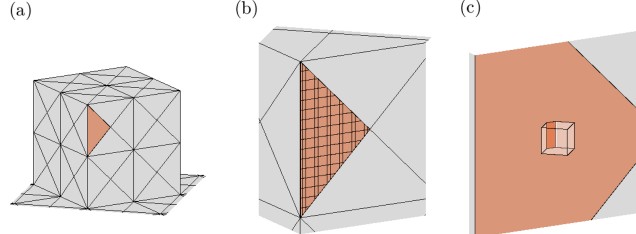

**Figure 6.** Surface treatment in uDALES v2.0: **(a)** triangular facets, with one coloured orange; **(b)** a facet divided into sections according to grid cells; **(c)** a facet section and fluid boundary cell.

An alternative approach would be to enforce that the value of a variable on the boundary, as determined by interpolating the values at nearby points, satisfies the boundary condition exactly. For example, one method would be to set the values at solid boundary points such that when interpolated across the boundary in the facet normal direction to an image point inside the fluid region, the value or flux on the boundary itself satisfies the desired flow condition (e.g. Majumdar et al., 2020). This approach is often used with DNS since the flow is fully resolved, which is not always the case for LES. Another inherent challenge with this approach is that it is more complicated to ensure global conservation. In the current approach, this is guaranteed since local conservation is ensured by enforcing zero flux (due to advection and diffusion) between each fluid–solid pair. In the alternative approach, the zero flux condition is at the boundary itself in an interpolated sense so local conservation no longer necessarily holds between fluid–solid pairs, but global conservation would still need to be enforced.

### 3.3.2 Surface fluxes

The IBM in uDALES uses wall functions to determine surface fluxes of momentum, temperature, and humidity at the fluid boundary points. Given the IBM is conservative, these fluxes are the only mechanism by which the surface exerts skin friction and exchanges heat and moisture with the fluid. The novelty of uDALES v2.0 is in the treatment of these fluxes given the surface is not aligned with the grid. Another conservation principle that motivates the approach is that for the surface energy balance model, the total heat flux into the fluid is equal to the total heat flux out of the surface. Whilst this principle is not relevant for momentum flux, the same method is also applied for that calculation.

While many IBMs with parameterised fluxes are effectively from the perspective of the fluid boundary points (Ma and Liu, 2017; Bao et al., 2018), the approach described here is based on considering the effect of each and every part of the surface on the fluid. As depicted in Fig. 6, the surface is divided cell-wise (for each grid) into facet sections, such that each section lies in only one cell. Note that this cell can be fluid or solid – in either case, the flux it imparts to the fluid must be accounted for. In this approach, each section imparts flux to a single fluid boundary point, which means it is necessary to identify the fluid boundary point that is most appropriate to experience the flux. This process is illustrated in Fig. 7. Note that the wall functions are defined in a wall-normal direction, and in this approach, the relevant "wall" is the facet section. Therefore, in many cases, the chosen fluid boundary point for each facet section is the one for which there exists a vector from the facet section to the fluid boundary point in the direction of the facet normal. However, there may be some facet sections that do not have a fluid boundary point lying in the direction of the facet normal. For such facet sections, the appropriate fluid boundary point (to impart the flux to) is chosen to be the point which maximises $\cos\theta/d$, where $\theta$ is the angle between the facet normal and the surface-point vector, and $d$ is the minimum distance between the facet section and the point. This minimises the angle between the surface normal and the surface-point vector as well as the distance to the point, because a closer point feels the effect of the surface to a greater extent.

For each facet section, the flux imparted to the appropriate fluid boundary point is calculated according to the wall functions using the properties of the facet that the section makes up, as well as the velocity and scalar fields, as discussed in Sect. 2.2. If $d$ is sufficiently large, the fields are evaluated at the fluid boundary point itself, whereas if $d$ is small then they are evaluated at a reconstruction point further from the surface, located at the boundary with the adjacent cell in the normal direction. This is because the wall functions are not valid when $d$ is small. The reconstruction approach is similar to that of Ma and Liu (2017); further detail on when and how it is used is provided in Appendix B.

A local orthonormal coordinate system is defined (at the location where the field is evaluated) using the facet normal $\hat{\boldsymbol{n}}$; the spanwise direction $\hat{\boldsymbol{p}} = (\hat{\boldsymbol{n}} \times \boldsymbol{u})/||\hat{\boldsymbol{n}} \times \boldsymbol{u}||$, where $\boldsymbol{u}$ is the velocity; and the streamwise direction $\hat{\boldsymbol{t}} = \hat{\boldsymbol{p}} \times \hat{\boldsymbol{n}}$. The streamwise velocity component $\boldsymbol{u} \cdot \hat{\boldsymbol{t}}$ corresponds to the velocity $u_{\mathrm{a}}$ used in the wall functions.

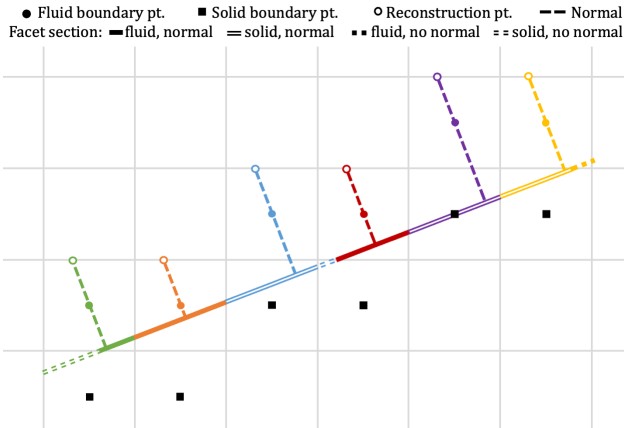

● Fluid boundary pt.    ■ Solid boundary pt.    ○ Reconstruction pt.    ‒ ‒ Normal
Facet section: ▬ fluid, normal   ▬ solid, normal   ▪▪ fluid, no normal   ▪▪ solid, no normal

**Figure 7.** A 2D facet divided into sections. Fluid boundary points (and reconstruction points) are coloured uniquely. Facet sections are coloured by their corresponding fluid boundary point and distinguished by whether they are in a fluid or solid cell and whether a normal vector exists between the section and fluid boundary point.

The surface stress (calculated using Eq. 3) is a tensor that must be transformed back to the corresponding Cartesian direction. This transformation is simplified because in the local basis, the stress is entirely parallel to $\hat{t}$ and perpendicular to $\hat{n}$. The global basis is denoted by $e_i$, where $i = 1, 2$, and $3$ corresponds to $\hat{x}$, $\hat{y}$, and $\hat{z}$, respectively, and the local basis by $e'_i$, where $i = 1, 2$, and $3$ corresponds to $\hat{t}$, $\hat{p}$, and $\hat{n}$, respectively. The local stress tensor $\tau'$ is therefore defined by $\tau'_{13} = \tau_w$ and $\tau'_{ij} = 0$ otherwise. The transformation to the global basis is given by

$$\tau_{ij} = \sum_l \sum_m (e_i \cdot e'_l)(e_j \cdot e'_m)\tau'_{lm}. \tag{8}$$

The total stress in direction $i$ is equal to $||\tau \cdot e_i||$, and the volumetric sink of momentum at the fluid boundary point $S_{u_i} = ||\tau \cdot e_i||a/(\rho V)$, where $a$ is the facet section area and $V$ is the cell volume.

The potential temperature and specific humidity fluxes are calculated using Eqs. (4) and (5), respectively, and correspond to source and sink terms $S_{\theta_l} = u_\tau \theta_\tau a/V$ and $S_{q_t} = u_\tau q_\tau a/V$. If the surface energy balance model is in use, the sensible and latent heat fluxes for facet $f$ are given by

$$H_f = \frac{\rho c_p}{A_f} \sum_{n \in C_f} a_n (u_\tau \theta_\tau)_n, \quad E_f = \frac{\rho L_v}{A_f} \sum_{n \in C_f} a_n (u_\tau q_\tau)_n. \tag{9}$$

Here, $C_f$ is the set of facet sections comprising facet $f$, and $A_f = \sum_{n \in C_f} a_n$ is the facet area. Given this formulation, global conservation with the surface energy balance is guaranteed. This is because each section is accounted for exactly once (and the sum of their area equals the total surface area), and for each section the fluid cell and facet experience an equivalent flux determined by the section area.

Note that the computational performance of the IBM scales linearly with the total number of facet *sections*, rather than the number of facets, and the number of sections depends on the total surface area of the geometry and the grid resolution. However, the number, or equivalently the size, of facets should be appropriate for the resolution (and vice versa). If the facets are similar in size to the grid cells, this may cause problems with accuracy, because the issues relating to distance ($d$) tend to be more pronounced with relatively small facets.

### 3.4 Shortwave radiation

The net shortwave radiation on a single facet ($K_f^*$) is the difference between the incoming ($K_f^\downarrow$) and outgoing ($K_f^\uparrow$) shortwave radiation, i.e. $K_f^* = K_f^\downarrow - K_f^\uparrow$. According to the radiosity assumption, transmission is neglected and all reflection is diffuse, meaning $K_f^\uparrow = \alpha_f K_f^\downarrow$, where $\alpha_f$ is the facet albedo. Therefore, the key quantity is $K_f^\downarrow$, which is the sum of the direct solar ($S_f$), diffuse sky ($D_f$), and reflected ($R_f$) components, i.e. $K_f^\downarrow = S_f + D_f + R_f$. Note that these quantities have units of $\mathrm{W\,m^{-2}}$.

The direct solar radiation depends on the position of the sun, which is specified using the solar azimuth ($\Omega$) and zenith ($\Theta$). For a coordinate system defined such that the $x$ direction is at an angle $\Omega_x$ clockwise from north, letting $\Omega' = \Omega - \Omega_x$, the vector to the sun $\hat{s}$ (referred to as the sun vector) is defined by

$$\hat{s} = \begin{bmatrix} \sin\Theta\cos\Omega' \\ -\sin\Theta\sin\Omega' \\ \cos\Theta \end{bmatrix}. \tag{10}$$

In uDALES v1.0, $S_f$ is calculated according to $S_f = I\hat{n}_f \cdot \hat{s}$, where $I$ is the direct normal irradiance (a model parameter) and $\hat{n}_f$ is the facet normal. To account for shading, rays are projected from the facet corners and centre in the direction of the sun vector to check for intersection with any other facet. If so, then shading occurs, and $S_f$ is reduced by a fraction for each of these points for which there is an intersection. This is a crude approximation for quantifying the radiation on shaded facets and scales poorly as $\mathcal{O}(N^2)$, where $N$ is the number of facets.

The method used to calculate direct solar radiation in uDALES v2.0 is designed to address these limitations. It involves projecting the facets onto a plane whose normal is parallel to the sun vector. To define a point on this plane $p_0$, take the centre of the geometry $p_g = [L_x/2\ L_y/2\ 0]^\top$ and project some reasonable distance $L$ in the direction of the sun vector: $p_0 = p_g + L\hat{s}$. To define an orthonormal pair of vectors spanning the plane, let $\hat{q} = [-\hat{s}_2\ \hat{s}_1\ 0]^\top$ and $\hat{r} = \hat{q} \times \hat{s}$. Given a point $p$, its projected position onto the plane is equal to $p - \mu\hat{s}$ for some unknown $\mu$, which is the distance between the point and the plane. The projected position can also be

written as $\boldsymbol{p}_0 + \zeta\hat{\boldsymbol{q}} + \eta\hat{\boldsymbol{r}}$ for some unknown $\zeta$ and $\eta$, which can be thought of local coordinates with respect to the origin $\boldsymbol{p}_0$. Equating these yields the system of equations for the unknowns:

$$\boldsymbol{p} - \boldsymbol{p}_0 = \begin{bmatrix} \hat{\boldsymbol{q}} & \hat{\boldsymbol{r}} & \hat{\boldsymbol{s}} \end{bmatrix} \begin{bmatrix} \zeta \\ \eta \\ \mu \end{bmatrix}. \tag{11}$$

Solving this system for all the vertices making up the triangulated surface yields the projected position of all the facets in terms of $\zeta$ and $\eta$. The projected facets potentially overlap, which correspond to one blocking the path of radiation to the other, i.e. shading. The cell centres are therefore also projected so that $\mu$ can be used to sort the facets by distance from the plane. The plane is discretised into pixels, which are classified according to whether they are inside or outside a projected facet using a modified version of the MATLAB function poly2mask (which has also been ported to Fortran for improved performance). This classification is carried out sequentially for all the facets in order of decreasing distance to the plane so that in the case of overlaps, the pixels are recorded as belonging to the closer facet. The result is effectively an image of the facets from the sun's point of view, with pixels labelled according to which facet they correspond to. The unshaded projected area of a facet ($A'_f$) is calculated by summing the pixels labelled by it and scaling the result by the discretisation resolution. Finally, denoting the actual facet area by $A_f$, then $S_f = I A'_f / A_f$, which is equal to $I \hat{\boldsymbol{n}}_f \cdot \hat{\boldsymbol{s}}$ for unshaded facets. This formulation accurately captures the radiation on shaded facets and scales as $\mathcal{O}(N)$ because there are no pairwise facet calculations. However, note that obtaining an accurate result is dependent on discretising the plane at a sufficiently high resolution with respect to the size of the facets.

Once the direct solar radiation on each facet has been obtained, the net shortwave radiation is calculated via the same conceptual procedure as in uDALES v1.0, because it is not dependent on the geometry representation. The diffuse sky radiation on each facet is given by $D_f = \psi_{f,\text{sky}} D_{\text{sky}}$, where $D_{\text{sky}}$ is a model parameter, and $\psi_{f,\text{sky}}$ is the sky view factor for facet $f$, a purely geometrical quantity defined as the fraction of diffuse radiation that leaves facet $f$ and does not impinge on any other facet. For calculation details of $\psi_{f,\text{sky}}$, see Suter et al. (2022). The reflected component on facet $f$ is given by

$$R_f = \sum_{g=1}^{N} \psi_{f,g} K_g^{\uparrow} = \sum_{g=1}^{N} \psi_{f,g} \alpha_g K_g^{\downarrow}$$
$$= \sum_{g=1}^{N} \psi_{f,g} \alpha_g (S_g + D_g + R_g), \tag{12}$$

where the view factor $\psi_{f,g}$ is the fraction of diffuse radiation leaving facet $f$ that impinges on facet $g$. Equation (12) shows that the reflected component on each facet depends on

that of all the other facets. This linear system of equations is solved iteratively to avoid the inversion of a large matrix (Suter et al., 2022).

In uDALES v1.0, view factors are calculated using a bespoke algorithm implemented in MATLAB (Suter, 2018). This was designed for rectangular facets and would be challenging to modify to support triangular facets. In uDALES v2.0, view factors are calculated using the open-source program View3D (DeGraw and Walton, 2018), which is written in the C programming language and supports triangular facets. This offers better performance due to the optimisation techniques it employs and the fact that C tends to have a faster runtime than MATLAB. Also, note that the shortwave radiation calculation occurs as a pre-processing step, meaning it does not affect the performance of simulations in the sense discussed in Sect. 3.1.

### 3.5 Periodic precursor simulations

Precursor simulations can be used to provide the time-dependent boundary conditions required to perform inflow–outflow simulations with LES (Suter et al., 2022; Grylls et al., 2019; Lim et al., 2022). They typically use periodic boundary conditions, and their advantage is that all spatial and temporal statistics of the eddies (e.g. two-point correlations) are representative of real turbulence provided the precursor simulation has an appropriate aspect ratio domain. This is in contrast to the common alternative approach of quantities being prescribed (on the inlet) and turbulence being generated synthetically (e.g. Xie and Castro, 2008). The precursor technique can be simpler to implement as it requires less prior information, though given the flow is strongly determined by the bottom boundary condition, trial and error with respect to the geometry is often required to obtain the desired turbulence statistics (if trying to match specific data). However, the streamwise and spanwise extent of the precursor simulation is typically limited, which can lead to the development of streamwise structures that span the entire domain and become permanent features of the average flow field. These so-called superstructures are observed in reality (Hutchins and Marusic, 2007) but are exacerbated by the limited domain size used in precursor simulations. This is undesirable because it results in spanwise variations of Reynolds-averaged quantities.

uDALES v1.0 simulations suffer from artificial superstructures as there is no functionality for counteracting their formation. In contrast, uDALES v2.0 has a technique designed to avoid them by applying an advection forcing term $\mathcal{F}_i = V(x, z)\frac{\partial u_i}{\partial y}$. Assuming mean flow in the $x$ direction only, the effect of the velocity $V(x, z)$ is to shift the flow by a spanwise displacement $\Delta_y$ over a streamwise distance

$\Delta_x$, i.e.

$$\Delta_y = \frac{1}{\langle u \rangle} \int_0^{\Delta_x} V(x,z) \mathrm{d}x, \tag{13}$$

where $\langle u \rangle(z)$ is the plane-averaged streamwise velocity. The velocity $V$ can be chosen arbitrarily, but in order to avoid discontinuities in the forcing field, we require that $V = 0$ at $x = 0$ and $x = \Delta_x$. Assuming a sinusoidal $x$ dependence that satisfies these conditions, i.e $V(x,z) = \hat{v}(z) \sin(\frac{\pi x}{\Delta_x})$ for some $\hat{v}(z)$ to be determined, the integral above can be evaluated with result

$$\Delta_y = \frac{2\Delta_x}{\pi} \frac{\hat{v}}{\langle u \rangle}. \tag{14}$$

Therefore, $\hat{v} = \frac{\pi \Delta_y}{2\Delta_x} \langle u \rangle$, which implies that the momentum forcing is given by

$$\mathcal{F}_i = \frac{\pi \Delta_y}{2\Delta_x} \langle u \rangle \sin\left(\frac{\pi x}{\Delta_x}\right) \frac{\partial u_i}{\partial y}, \tag{15}$$

where $\Delta_x$ and $\Delta_y$ are parameters that can be chosen freely. For the simulations in this text using the method, $\Delta_x = L_x/2$, $\Delta_y = L_y/4$, and the output plane is taken at $x = 0$.

## 4 Validation

### 4.1 Neutral flow over a staggered array of cubes

In this case, uDALES v1.0 and v2.0 are set up following a numerical simulation by Xie et al. (2008), which compares against an experiment by Cheng and Castro (2002). The set-up in uDALES is full scale; i.e. the dimensions are 1000 times larger than the experiment and numerical simulations presented in the mentioned literature. This case does not use any of the upgraded features of uDALES v2.0, and so the flow is expected to match that of v1.0.

The case consists of neutral flow around a staggered array of cubes with width and mean height equal to $h_m = 10$ m (Figs. 8a and b). Periodic lateral boundary conditions are used, with free slip at the top. The flow is forced by a constant kinematic pressure gradient $\mathcal{F}_x = 4.1912 \times 10^{-3}$ m s$^{-2}$. The domain size is $L_x \times L_y \times L_z = 16h_m \times 16h_m \times 10h_m$, and the number of grid cells is $N_x \times N_y \times N_z = 256 \times 256 \times 256$. The initial velocity profile is uniform; the value is equal to the vertically averaged profile of Xie et al. (2008), which was estimated to be $5.15$ m s$^{-1}$. Note that since the geometry is aligned with the grid and the lateral boundaries are periodic, the IBM and pressure solver used in both versions of uDALES are conceptually the same.

The spatially averaged mean streamwise velocities are presented in Fig. 8c, which shows that uDALES v1.0 and v2.0 are in good agreement with each other and with the literature.

Additionally, the streamwise mean and rms velocity profiles at different spanwise locations (at $y/h_m = 0, 1, 2, \ldots, 7$) along the centre line between the third and the fourth row of the buildings (i.e. at $x/h_m = 6$) are presented in Figs. 8d–f. Note that each profile here is computed as an average of the four available profiles within the computational domain following Xie et al. (2008). There is close agreement with the LES data of Xie et al. (2008), but the agreement with the experimental measurements is less good, particularly higher up in the flow domain. These differences are most likely for the same reasons discussed in Xie et al. (2008), namely that the experimental and numerical studies each use a different domain height. The results suggest that uDALES v2.0 can accurately simulate neutral flows over idealised, grid-aligned geometries with periodic boundary conditions, which is important because such cases are often used as precursor simulations (see Sect. 3.5).

### 4.2 Neutral cross-ventilation flow for isolated building

This case involves using uDALES v2.0 to simulate the cross-ventilation of an isolated enclosure (building) with rectangular openings (windows) in both windward and leeward walls (van Hooff et al., 2017). The results are compared with both the RANS and LES simulations of van Hooff et al. (2017) and the wind tunnel measurements of Tominaga and Blocken (2015). Note that uDALES v1.0 cannot handle a geometry of this kind, as it is not possible to have blocks overhanging fluid, and so this case is intended to validate the upgraded IBM for a novel application: coupled indoor–outdoor flow.

The interior volume of the enclosure has dimensions $D \times W \times H = 0.2$ m $\times 0.2$ m $\times 0.16$ m; see Fig. 9a and b. In contrast to the simulations in van Hooff et al. (2017) that were carried out using an unstructured mesh, the wall thickness cannot be specified independently of the grid resolution when using an IBM. At least three grid points must be present inside the solid domain, which translates to a wall thickness of $0.02$ m. The dimensions of the windows are $W_0 \times H_0 = 0.092$ m $\times 0.036$ m, and they are placed $0.062$ m from the bottom and $0.054$ m from the inner lateral walls. The computational domain size is $L_x \times L_y \times L_z = 3.12$ m $\times 1.84$ m $\times 0.96$ m, which means that the inlet and outlet boundaries are at distances of $3H$ and $15H$ from the windward and leeward walls of the building, respectively. The lateral sides and top boundary of the computational domain are at a distance of $5H$ from the building walls/roof. The number of grid cells is $N_x \times N_y \times N_z = 1024 \times 512 \times 128$, and the $z$ grid is stretched, with the finest grid resolution near the building being approximately $0.0035$ m in each direction. Inflow–outflow boundary conditions are used in the $x$ direction, while the $y$ direction is periodic and the top is free slip. A precursor simulation is used for the inflow to match the fully developed neutral boundary layer inlet used by van Hooff et al. (2017) such that the streamwise veloc-

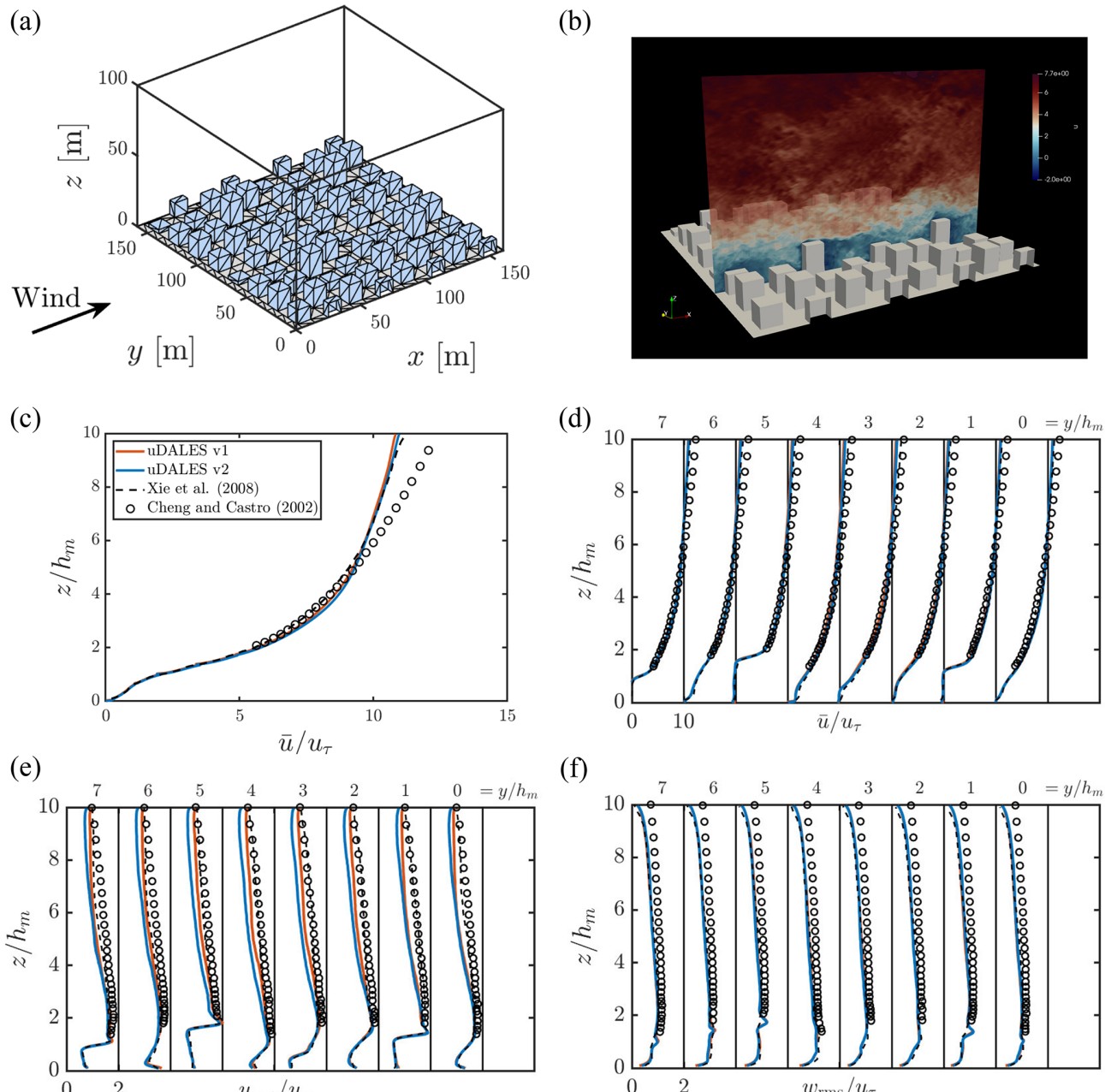

**Figure 8.** Staggered-cube case (Sect. 4.1): **(a)** computational domain; **(b)** instantaneous streamwise velocity contour at an arbitrary time instant; **(c)** spatially averaged mean streamwise velocity profile; **(d)** mean and **(e)** rms streamwise velocity and **(f)** rms vertical velocity profiles at different locations behind the third row of buildings. The legend shown in panel **(c)** is also true for all the panels.

ity at a height of $H = 0.16$ m equals to the reference velocity $U_\mathrm{H} = 4.3\,\mathrm{m\,s^{-1}}$.

Figure 9c shows a comparison of the streamwise velocity profile computed by uDALES v2.0 with standard $k$-$\varepsilon$ RANS (RANS|SKE) and LES results of van Hooff et al. (2017) and the experimental measurements of Tominaga and Blocken (2015), along three different vertical lines at $x/D = 0.25$, 0.5, and 0.75 in the vertical centre plane ($y/W = 0$) of the enclosure shown using dashed lines in Fig. 9b. A simi-

lar comparison for the turbulent kinetic energy (TKE) profile is also shown in Fig. 9d. Velocity and TKE along the horizontal centre line ($y/W = 0$, $z/H = 0.5$) are shown in Fig. 9e. All these figures demonstrate good agreement between uDALES v2.0 and those reported in the literature. Furthermore, uDALES v2.0 captures the shape and width of the jet entering through the window accurately as shown in terms of the dimensionless mean velocity magnitude ($|V|/U_\mathrm{H}$) contour in Fig. 9f. At $x/D = 0.625$, uDALES v2.0 predicts

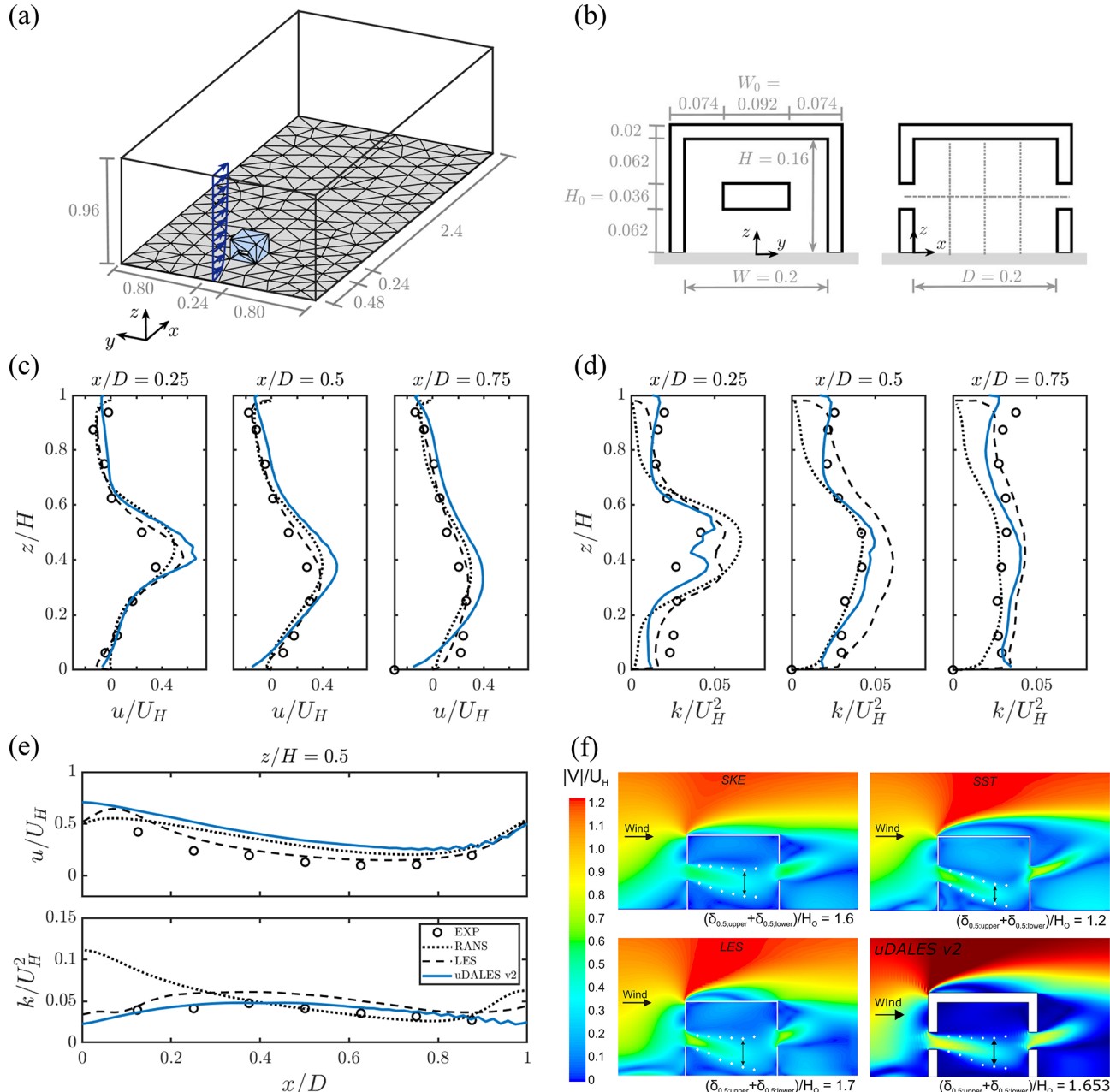

**Figure 9.** Cross-ventilation case (Sect. 4.2): **(a)** computational domain; **(b)** front and side view of the building, including windows (all dimensions are in m). **(c)** Mean streamwise velocity, and **(d)** mean turbulent kinetic energy (TKE) profiles at streamwise locations inside the building at the vertical centre plane ($y/W = 0$). **(e)** Mean streamwise velocity and TKE profiles along a horizontal line through the middle of the windows ($y/W = 0, z/H = 0.5$). The legend shown in bottom subpanel of panel **(e)** holds for all the panels in all frames. **(f)** Contours of dimensionless mean velocity magnitude ($|V|/U_H$; SKE, SST and LES panels are reproduced here from van Hooff et al. (2017) with appropriate permission from the publisher.

the jet width $(\delta_{0.5;\text{upper}} + \delta_{0.5;\text{lower}})/H_0 = 1.653$ to fall between the jet width prediction by RANS|SKE ($= 1.6$) and LES ($= 1.7$) simulations of van Hooff et al. (2017). Here, $\delta_{0.5}$ indicates the jet half width – the vertical distance between where the jet velocity magnitude is a local maximum and where it is equal to half the maximum at a particular

$x$ location. These results demonstrate that uDALES v2.0 can be used to simulate coupled indoor–outdoor flows. However, this is a relatively simple case; a multi-physics validation including heat transfer or pollutant dispersion would be more challenging, and few validation studies currently exist (e.g. Kosutova et al., 2024).

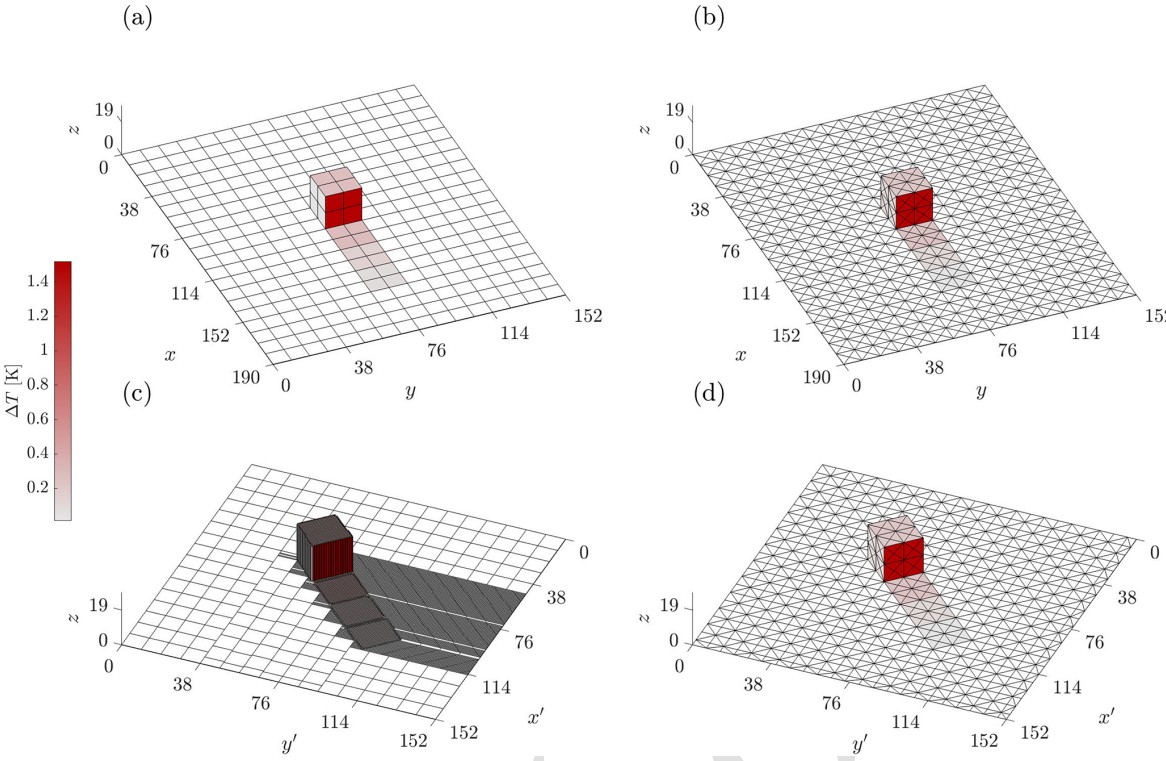

**Figure 10.** Single-cube case with constant surface temperature (Sect. 4.3): facets coloured according to temperature difference $\Delta T$ for **(a)** v1.0, original; **(b)** v2.0, original; **(c)** v1.0, rotated; and **(d)** v2.0, rotated.

## 4.3 Non-neutral flow over a single cube with surfaces at constant temperature

This case is based on an LES study by Boppana et al. (2013), in which the flow is compared against an experiment conducted by Richards et al. (2006). Here both uDALES v1.0 and v2.0 are used, and the simulations are run at full scale, i.e. 100 times larger than in the mentioned literature. The geometry consists of a single cube, and a subset of the facets are heated to a constant temperature, hence the results are compared to a non-neutral case presented in Boppana et al. (2013). In the original case (Fig. 10a and b) the cube is grid-aligned, but this study also includes simulations for which the cube is rotated with respect to the grid (Fig. 10c and d). For each simulation, the intention is to maintain the same physical flow over the cube. Figure 10c shows the effect of using a voxel representation for the geometry – the facets for the uDALES v1.0 rotated case need to be much smaller (and more numerous) in places where the actual geometry does not align with the grid. This case is intended to test the ability of the upgraded IBM to handle non-grid-aligned geometries, particularly with respect to modelling heat transfer (without using the surface energy balance scheme). Note that is also possible to specify a constant heat flux in uDALES, but this option is not tested in the current work as it does not involve the wall functions. The simulations use inflow–outflow

boundary conditions, thereby testing another upgraded feature of uDALES v2.0

The cube has a side length of $h = 19$ m, and in the original (non-rotated) case the domain size is $L_x \times L_y \times L_z = 10h \times 8h \times 6h$. Note that the domain width has been increased from that used by Boppana et al. (2013) so that it matches that of the rotated case; therefore, the two cases can use the same precursor simulation. The centre of the cube is located at $x_c = 4.5h$, $y_c = 4h$. For the rotated case, the flow is rotated by an angle of $45°$, and the domain size is $L'_x = L'_y = 8h$. The cube is positioned such that the distance from the origin to the cube centre is $4.5h$; i.e. the centre is at $x'_c = 4.5h/\sqrt{2}$, $y'_c = 4.5h/\sqrt{2}$. This means that the distance from the cube centre to the outlet is at least $5.5h$, thus making it possible to compare the wake with the original case. The grid sizes of the original and rotated cases are $N_x \times N_y \times N_z = 320 \times 256 \times 192$ and $N'_x \times N'_y \times N'_z = 256 \times 256 \times 192$, respectively, meaning the resolution is 0.59375 m in each direction.

The temperature of the facets is set to match the Richardson number ($Ri$) of Boppana et al. (2013), which requires setting the temperature differences between the facets and the air to be 100 times smaller (as shown in Table 2) CE1. CE2 The cases corresponding to $Ri = 0$ (neutral) and $Ri = -0.66$ (non-neutral) were run, and the non-neutral results are presented here. The Reynolds number ($Re$) for urban flows is typically sufficiently large for the viscous stresses to be negli-

**Table 2.** Single-cube case with constant surface temperature (Sect. 4.3): temperature difference ($\Delta T$) between facets and the inlet.

|  | Lee | Roof | Side | Floor 1 | Floor 2 | Floor 3 |
|---|---|---|---|---|---|---|
| $\Delta T$ [K] | 1.52 | 0.26 | 0.14 | 0.28 | 0.14 | 0.07 |

gible compared to the turbulent stresses and pressure; i.e. the flow is essentially independent of $Re$. This means the geometry scale can be changed while keeping flow velocity constant (and thus changing $Re$) and obtain similar results, as in Sect. 4.1. However, $Ri$ is a measure of the relative strength of buoyancy to the mean kinetic energy, and this ratio must stay the same; otherwise, the character of the flow will change. It is not clear what values to set for the momentum and heat roughness lengths, as a different wall model is used in Boppana et al. (2013). Here $z_0 = 0.01$ m is used, with $z_{0,h} = z_0$ as in Cai et al. (2008).

Regarding boundary conditions, the $y$ direction is periodic and the vertical velocity at the top is allowed to vary as discussed in Sect. 3.2. The $x$ direction is inflow–outflow, with the inflow provided by a precursor simulation, whereas Boppana et al. (2013) used synthetic turbulence generation. The geometry consists of staggered cubes of side length $h/8$ and plan area fraction $\lambda_p = 0.25$. The domain length is $12h$, with the same width, height, and resolution as the main simulation. The aim of the precursor is to generate mean streamwise velocity and turbulent kinetic energy profiles that match that of Richards et al. (2006) and Boppana et al. (2013). In the experiment, the flow is characterised by a power-law profile with exponent $\alpha = 0.52$:

$$u_{\mathrm{exp}}(z) = u_{\mathrm{ref}}\left(\frac{z}{z_{\mathrm{ref}}}\right)^{\alpha}. \qquad (16)$$

The reference height $z_{\mathrm{ref}}$ was taken to be 10 m, and the reference velocity $u_{\mathrm{ref}}$ was estimated to be $0.33\,\mathrm{m\,s^{-1}}$. The initial wind speed is set to this profile, and the plane-averaged velocity is forced towards it by using a nudging term:

$$\mathcal{F}_x = \frac{u_{\mathrm{exp}} - \langle u \rangle}{t_{\mathrm{nudge}}}, \qquad (17)$$

with $t_{\mathrm{nudge}} = 60$ s. In order to avoid the formation of artificial superstructures, the shifting method described in Sect. 3.5 is used. Figure 11c shows the time-averaged inflow plane velocity; it is reasonably uniform in the $y$ direction, indicating that the shifting method is successful. Averaging this plane in the $y$ direction results in the velocity profile shown in Fig. 11a, which matches the power-law profile almost exactly. The TKE is shown in Fig. 11b. The profile broadly agrees with the experimental and numerical values, in that the peak is at around 0.02 between $z/h = 1$ and $z/h = 3$ and decreases above this height.

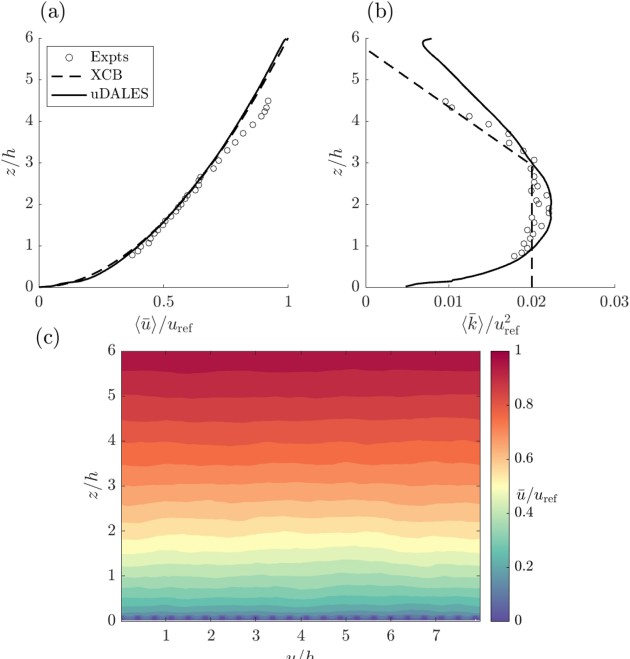

**Figure 11.** Single-cube case with constant surface temperature (Sect. 4.3): spanwise-averaged inflow profiles of **(a)** wind speed and **(b)** turbulent kinetic energy; **(c)** spanwise dependence of wind speed. In the legend, "Expts" refers to Richards et al. (2006) and "XCB" refers to Boppana et al. (2013).

For the main simulation, Fig. 12c–f shows slices of mean scaled potential temperature halfway up the cube. From this it appears that the v2.0 rotated case agrees better with the original v2.0 case, and indeed the original v1.0 case, than the v1.0 rotated case does. The slight asymmetry in the wake in the rotated cases is most likely due to the boundary conditions. The $y$ direction is periodic so that the turbulent inflow boundary condition in the $x$ direction can eventually reach both lateral sides of the cube, but the effective distance from the inlet to each of the sides is not the same. Figure 12a and b, respectively, show profiles of mean normalised streamwise velocity and potential temperature downstream of the cube on the wake centre line. The velocity profiles are all generally in agreement, with greater discrepancy below cube height as the distance from the wall increases. This shows that the models capture the wake similarly in the near field. The temperature profile at $\chi/h = 0.55$ is very close to the leeward face so the temperature gradient in the streamwise direction is large, making the comparison at this point less precise. Other than at this location, uDALES generally agrees well with the simulations of Boppana et al. (2013), with v2.0 matching more closely than v1.0 in the rotated case. Note that there is not good agreement with the experiments; this is discussed in detail in Boppana et al. (2013) and relates to the difficulty in determining the true boundary condition for heat on the leeward face. The results demonstrate

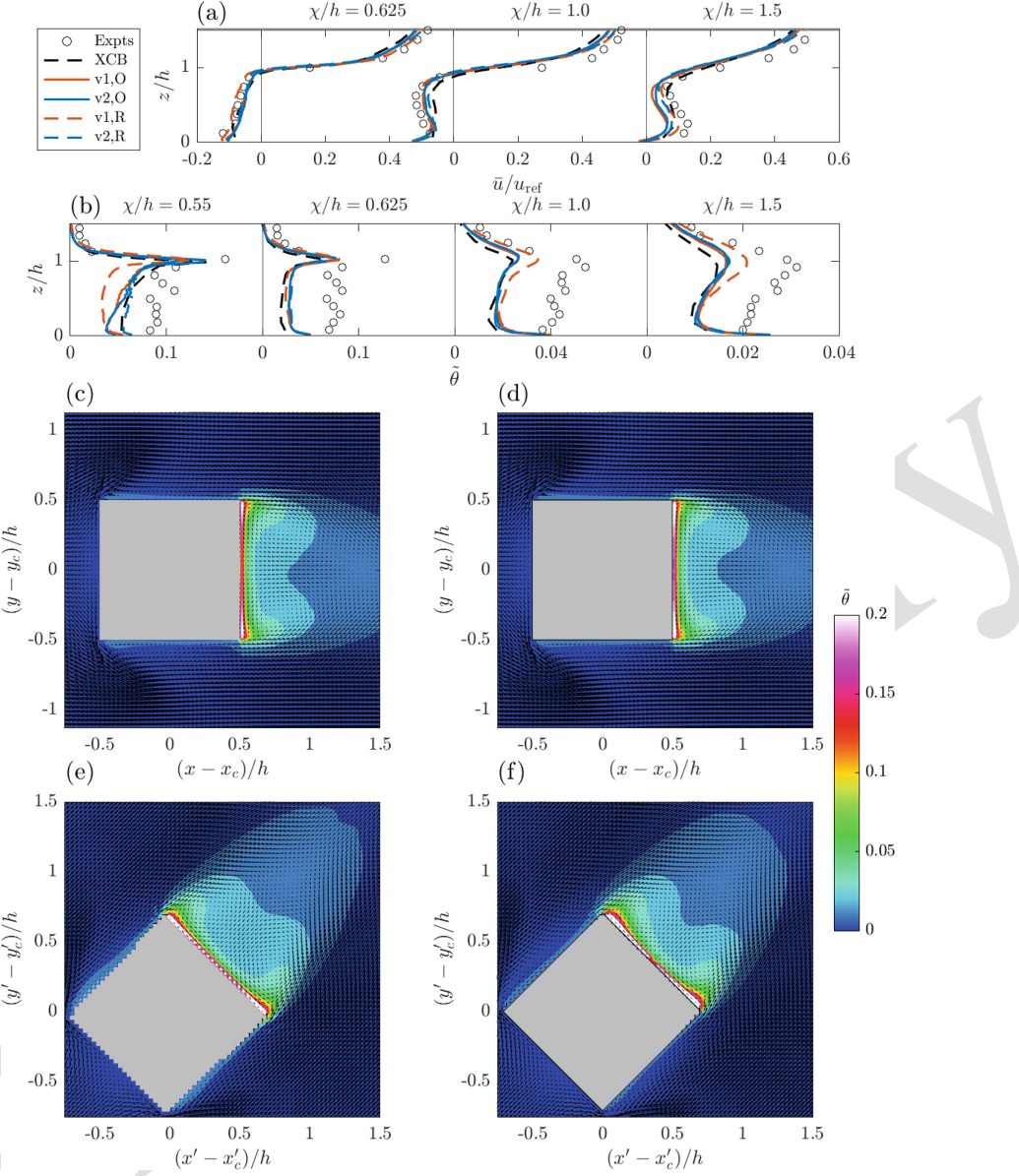

**Figure 12.** Single-cube case with constant surface temperature (Sect. 4.3): normalised mean profiles of **(a)** streamwise velocity $\bar{u}/u_{\mathrm{ref}}$ and **(b)** potential temperature $\tilde{\theta} = (\bar{\theta} - \theta_{\mathrm{ref}})/(T_{\mathrm{lee}} - \theta_{\mathrm{ref}})$ at downstream locations in the cube wake CE3, with streamwise distance from cube centre denoted by $\chi$. In the legend, "Expts" refers to Richards et al. (2006), "XCB" refers to Boppana et al. (2013), and "O" denotes original and "R" denotes rotated cases for uDALES v1.0 and v2.0. Below: contours of $\tilde{\theta}$ with mean velocity vectors at $z/h = 0.5$ for **(c)** v1.0, original; **(d)** v2.0, original; **(e)** v1.0, rotated; and **(f)** v2.0, rotated.

that there is a slight advantage in terms of accuracy when using uDALES v2.0 in situations with constant surface temperature compared to v1.0. The choice of rotation angle (45°) is in practice the most favourable for obtaining a symmetrical wake, most likely due to the effect of the lateral boundary conditions. It is expected that walls at arbitrary angles to the grid would perform equally well in cases where the lateral boundary conditions have a weaker influence.

## 4.4 Non-neutral flow over a single cube using the surface energy balance scheme

Similarly to Sect. 4.3, this case involves simulating the flow around the original (non-rotated) and rotated geometries using uDALES v1.0 and v2.0, though here the facet temperatures evolve according to the surface energy balance model. This is in order to validate the implementation in uDALES v2.0 against v1.0 in the situations where the two models

**Table 3.** Single-cube case with surface energy balance (Sect. 4.4): surface energy balance parameters.

| Parameter | Value |
|---|---|
| Solar azimuth ($\Omega$) | 90° |
| Solar zenith ($\Theta$) | 45° |
| Direct normal solar irradiance ($I$) | 800 W m$^{-2}$ |
| Diffuse sky shortwave irradiance ($D_{sky}$) | 80 W m$^{-2}$ |
| Diffuse sky longwave irradiance ($L_{sky}$) | 336 W m$^{-2}$ |
| Initial exterior facet temperature ($T_s$) | 293 K |
| Interior facet temperature ($T_B$) | 293 K |
| Facet albedo ($\alpha$) | 0.5 |
| Facet emissivity ($\epsilon$) | 0.85 |
| Facet volumetric heat capacity ($C$) | 0.1 MJ m$^{-3}$ K$^{-1}$ |
| Facet conductivity ($\lambda$) | 1 W m$^{-1}$ K$^{-1}$ |
| Facet thickness ($D$) | 0.4 m |
| Number of facet layers | 5 |

should agree and to evaluate how the rotated cases compare to the original cases.

The boundary conditions, including the precursor-defined inflow, are the same as in Sect. 4.3. The initial conditions are also unchanged apart from the facet temperatures. The simulation parameters pertaining to the surface energy balance are given in Table 3. Note the relatively low facet volumetric heat capacity used, which allows the simulations to reach a steady state faster. The steady state reached does not depend on the volumetric heat capacity, and it is only the steady state (and not transient behaviour) that is of interest here. The solar position is chosen such that of the cube faces, only the leeward and roof (should) receive direct solar radiation. This means these faces should be heated more than the others, so the flow is expected to be qualitatively similar to Sect. 4.3, meaning any considerations discussed there also apply here.

Figure 13 shows the steady-state sensible heat flux ($H$) on the facets. It is clear that the v1.0 rotated case differs both qualitatively and quantitatively. For example, on the leeward face, $H$ does not reach as high a value, and the shaded region (with low $H$) is not captured accurately. This latter observation relates to the fact that how exactly the geometry is discretised into facets is significant. In the shaded region, the facets are large in both dimensions, so the result would likely be improved by using much smaller facets. The facets making up the ground just downstream of the leeward face are very thin in the $x'$ direction but large in the $y'$ direction, meaning the structure (namely $H$ decreasing away from the cube) is not symmetric along the cube centre line like the other cases. This discussion indicates that the geometry specification for v1.0 is important, and this is ultimately because it is dependent on the underlying computational grid. Although the v2.0 rotated case is better, there is still some discrepancy, chiefly that the structure is not as symmetric along the cube centre line when compared to the original case. As discussed in Sect. 4.3, it is most likely that this discrepancy

is due to the boundary conditions – the $x'$ direction is inflow–outflow and the $y'$ direction is periodic, which allows the turbulent inflow to reach both sides of the cube but having effectively travelled different distances to each. Another factor in this study is that the incoming flow (with uniform temperature) will be affected by the warmer ground facets, causing thermal inlet effects and variation in the $x/x'$ direction. This is unavoidable when using the precursor simulation, but it seems from Fig. 13 that this effect has diminished sufficiently before reaching the cube so that the four cases are comparable.

Figure 14 show the steady-state surface heat fluxes and internal temperatures for the roof, leeward, and lateral faces of the cube. The temperature difference between the surface of the leeward face and $\theta_{ref}$CE4 is approximately 16 times higher than in Sect. 4.3, thus yielding $Ri \approx -10$. This indicates that buoyancy is more significant for this case. For the roof, all cases agree, as the effect of rotation is negligible. For the leeward and lateral sides, the v1.0 rotated case differs from the non-rotated cases to a much greater extent than the v2.0 rotated case; for the leeward side, the shortwave flux is around 21 % less. These differences are primarily caused by errors in the radiation, which is directly affected by the rotation. Note that the average of the direct shortwave radiation over the leeward face is in fact correct in the v1.0 rotated case. This is because although each facet receives less than the other cases as their normal is not aligned with the vector to the sun, the wall's surface area is larger by the same factor (equal to $\sqrt{2}$ for this choice of solar position and rotation angle of 45°). This result would also hold for other angles as long as there is no internal shading on the face, which clearly introduces errors in the direct contribution. The error is therefore due to the other contributions to the net shortwave radiation. The reflected contribution caused by neighbouring facets facing each other is incorrect because it obviously should be zero for a flat surface. However, the net shortwave on the leeward face is in fact less than expected, so this error must be outweighed by the error in the diffuse sky radiation. If neighbouring facets can "see" each other, their sky view factor is reduced, hence reducing this contribution. This discussion concerning contributions due to neighbour reflections and from the sky also holds for the lateral sides, though unlike the leeward face, the net shortwave on these sides is larger than expected. This is because the sides should not have a direct contribution as they are not visible to the sun, but this is not in fact true for the v1.0 rotated case. Around half of the facets are oriented such that they are visible to the sun, and these are not completely shaded by the neighbour that they face.

Regarding the other terms in the surface energy balance, the leeward and side faces have reduced incoming longwave radiation for the same reasons given for the shortwave. This is consistent with the fact that the net longwave is less negative, and the outgoing longwave is directly determined by surface temperature, which is lower in the v1.0 rotated case.

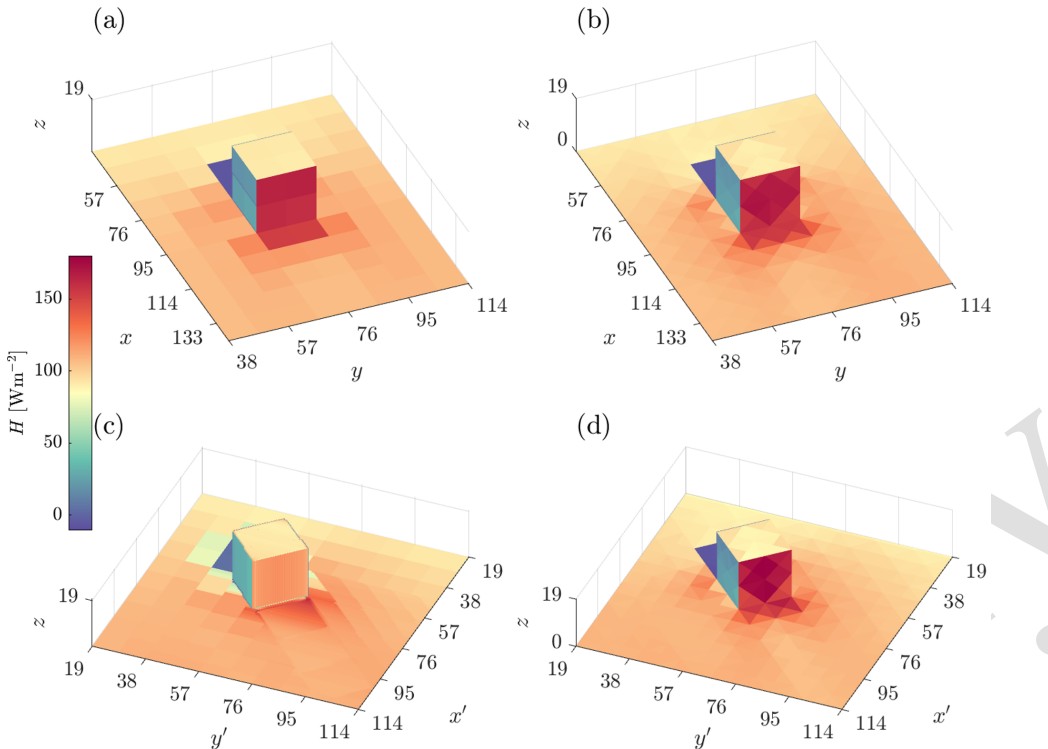

**Figure 13.** Single-cube case with surface energy balance (Sect. 4.4): facets coloured according to sensible heat flux $H$ for **(a)** v1.0, original; **(b)** v2.0, original; **(c)** v1.0, rotated; and **(d)** v2.0, rotated.

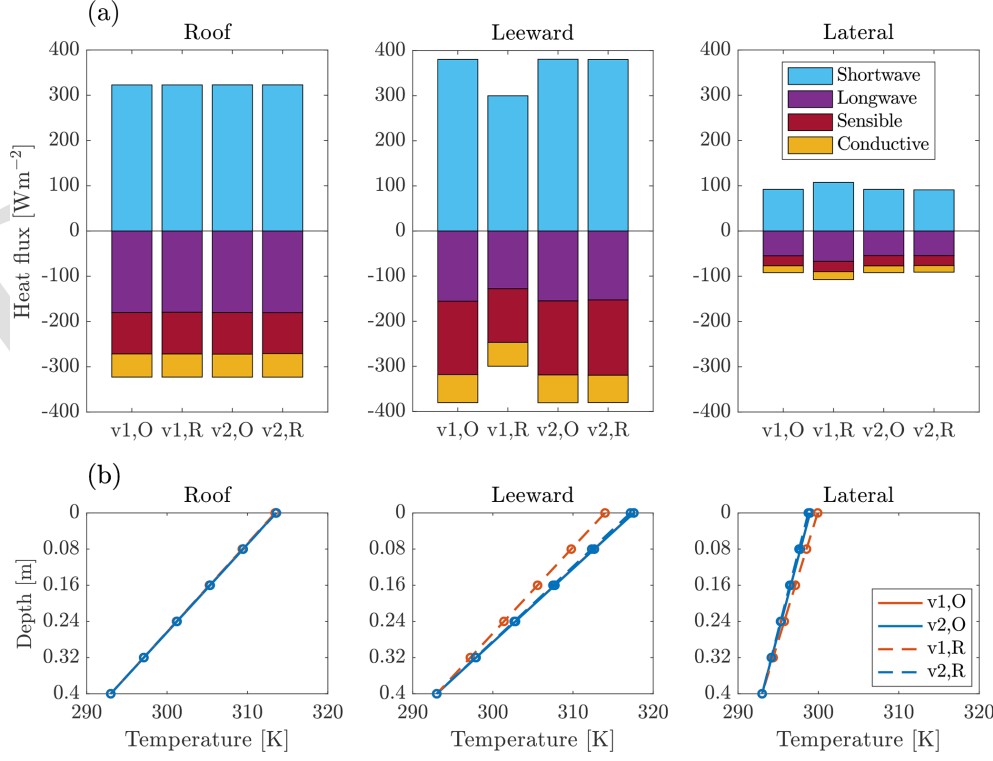

**Figure 14.** Single-cube case with surface energy balance (Sect. 4.4): steady-state **(a)** surface heat fluxes and **(b)** internal facet temperature for roof, leeward, and lateral facets.

The temperature of the leeward face is lower because it has less available energy due to the reduced shortwave, which also acts to reduce the sensible heat and conductive fluxes. The side faces have more available energy due to the increased shortwave (despite the reduced longwave), and so the temperature is higher, which acts to increase the other fluxes, though the effects are small. These results demonstrate that uDALES v2.0 is more accurate than v1.0 in capturing the radiative terms in the surface energy balance when the geometry not aligned with the grid, which in turn means the other terms are more accurate too. In this case, the errors associated with using uDALES v1.0 are due to the grid-aligned representation of each face of the cube individually and the non-physical radiative transfer that occurs as a result. For cases with more radiative interaction between facets, e.g. in street canyons, these errors would likely be compounded.

## 5  Conclusion

In this work, an upgrade to uDALES has been presented that addresses several key limitations with the previous version. The limit of parallelisation of the code has been increased by implementing a two-dimensional domain decomposition using the external library 2DECOMP&FFT, which was shown in Sect. 3.1. The pressure solver now makes use of the FFT algorithm for inflow–outflow boundary conditions, as was described in Sect. 3.2. Finally, the geometry representation has been improved in that it can be specified independently of the computational grid (using the widely used STL file format), and the IBM has been developed significantly to accommodate this, as described in Sect. 3.3. In particular, the novel approach taken to apply parameterised surface fluxes ensures that heat is conserved with the surface energy balance scheme. uDALES v2.0 has been compared against uDALES v1.0 and other physical and numerical simulations of neutral and non-neutral cases in order to validate the development. The neutral cases confirm that v2.0 produces very similar results to v1.0 in canonical urban cases (Sect. 4.1) and demonstrate its utility for coupled internal–external flows that are not possible to study using v1.0 (Sect. 4.2). The non-neutral cases (Sects. 4.3 and 4.4) were designed to evaluate both versions in situations where the geometry is not aligned with the grid, and the results demonstrate that v2.0 consistently exhibits better accuracy than v1.0.

This upgrade makes uDALES v2.0 a far more practical tool than v1.0. The code scales well for realistic problem sizes up to a large number of processors on ARCHER2, meaning simulations can be run far more quickly. This enables the study of larger urban areas and makes longer simulations, e.g. over a diurnal cycle, more feasible (e.g. Anders et al., 2023). The changes to the pressure solver mean that simulations can be performed just as quickly for inflow–outflow situations as for periodic ones, which was not the case for uDALES v1.0. The fact that non-grid-aligned obstacles can be modelled more accurately means that uDALES v2.0 can be used reliably in situations involving realistic urban geometries, which is important for wind loading and microclimate studies, for example. As a result, the capabilities of uDALES, and so of Cartesian LES codes more generally, are now positioned at the cutting edge of modelling techniques for urban environments and provide an attractive alternative to using LES codes that require body-fitted mesh generation.

This being said, some limitations of uDALES remain. The test cases presented here are simple by design, as the priority was to validate the fundamental aspects of the development (in particular the novel IBM). As such, the cases do not capture the full complexity of urban areas. In order to strengthen the robustness and applicability of the model, future work will involve evaluation of a more diverse and realistic range of urban morphology, materiality, and environmental conditions (e.g. Resler et al., 2021). In a future version, trees could be represented in the same way as the built environment (i.e. using a triangulated surface). This would require integrating trees into the 3D radiative transfer scheme, which is challenging for the current approach based on the radiosity assumption. Also, uDALES v2.0 still only has the option to specify a time-varying inlet in the $x$ direction (and not the $y$ direction), and this is obtained from a precursor simulation with the same resolution as the main simulation. Other models have the ability to nest a high-resolution simulation within a lower-resolution, larger-scale one, with the option of using a separate (mesoscale) model for the latter (Lin et al., 2021; Kadasch et al., 2021); this requires time-varying conditions on all lateral boundaries.

The work presented here provides a firm foundation for future development. The upgraded boundary condition implementation for inflow–outflow simulations in uDALES v2.0 does not fundamentally treat the $x$ and $y$ directions differently, meaning that it would be possible to implement a nesting technique. The new IBM approach is flexible to modification, such as incorporating interpolation in the way described in Sect. 3.3.1 or altering the exact form of the wall functions. Given the code now uses low-level routines from the 2DECOMP&FFT library, which is used by many other modelling tools, any development of them can potentially be adopted in uDALES. For example, the library supports parallel I/O via either MPI-IO or ADIOS2 backends (Rolfo et al., 2023). The ADIOS2 backend provides access to a streaming interface which can be used to implement in situ analysis, reducing the amount of data written to disc (Bartholomew et al., 2023). Also, support for NVIDIA GPUs has recently been added to 2DECOMP&FFT, with support for the other vendors planned (Rolfo et al., 2023), which will allow future efforts to port uDALES to GPUs to focus on the uDALES-specific components only. The use of GPUs would also suit a more advanced radiative transfer scheme that is based on ray tracing, which could capture specular reflection and trans-

mission, and is an attractive approach for modelling trees. In general, the improved low-level functionality of the codebase prepares uDALES for exascale HPC systems, which means that it will be increasingly feasible to perform ever-higher-resolution simulations of larger, more realistic urban areas, which is crucial to furthering understanding of urban climate processes.

## Appendix A: Pressure solver

Following Ferziger et al. (2020), the governing equations for velocity can be written (using an Euler time integration scheme for simplicity):

$$\frac{\boldsymbol{u}^{n+1} - \boldsymbol{u}^n}{\Delta t} = -\nabla p^{n+1} + \boldsymbol{r}^n. \tag{A1}$$

Here $\boldsymbol{r}^n$ includes all processes other than pressure. The pressure $p^{n+1}$ is determined using a two-step scheme which involves introducing the pressure correction $\pi^n = p^{n+1} - p^n$ and a predicted velocity $\boldsymbol{u}^*$ defined according to

$$\frac{\boldsymbol{u}^* - \boldsymbol{u}^n}{\Delta t} = -\nabla p^n + \boldsymbol{r}^n. \tag{A2}$$

Substitution into Eq. (A1) yields

$$\frac{\boldsymbol{u}^{n+1} - \boldsymbol{u}^*}{\Delta t} = -\nabla \pi^n. \tag{A3}$$

Requiring incompressibility $(\nabla \cdot \boldsymbol{u}^{n+1} = 0)$ results in a Poisson equation for the pressure correction:

$$\nabla^2 \pi^n = \frac{\nabla \cdot \boldsymbol{u}^*}{\Delta t}. \tag{A4}$$

Discretising in space and denoting the right-hand side of Eq. (A4) as $f$,

$$\frac{\pi_{i+1,j,k} - 2\pi_{i,j,k} + \pi_{i-1,j,k}}{\Delta x^2}$$
$$+ \frac{\pi_{i,j+1,k} - 2\pi_{i,j,k} + \pi_{i,j-1,k}}{\Delta y^2}$$
$$+ \frac{\frac{\pi_{i,j,k+1} - \pi_{i,j,k}}{\Delta z_{k+1/2}} - \frac{\pi_{i,j,k} - \pi_{i,j,k-1}}{\Delta z_{k-1/2}}}{\Delta z_k} = f_{i,j,k}. \tag{A5}$$

Next, take the DFT/DCT transform of both sides with respect to $x$ and then $y$, denote the modes using $n$ and $m$, respectively, and use hat notation to denote a transformed quantity.

$$a_k \hat{\pi}_{n,m,k-1} + b_{n,m,k} \hat{\pi}_{n,m,k} + c_k \hat{\pi}_{n,m,k+1} = \hat{f}_{n,m,k} \tag{A6}$$

Here $a_k = \frac{1}{\Delta z_k \Delta z_{k-1/2}}$, $c_k = \frac{1}{\Delta z_k \Delta z_{k+1/2}}$, and $b_{n,m,k} = -(a_k + c_k) + \lambda_n + \lambda_m$, where $\lambda_n$ and $\lambda_m$ are as follows:

$$\lambda_n = \begin{cases} \frac{2}{\Delta x^2}(\cos\frac{2\pi n}{N_x} - 1) & \text{DFT} \\ \frac{2}{\Delta x^2}(\cos\frac{\pi n}{N_x} - 1) & \text{DCT} \end{cases} \quad n = 0, \ldots, N_x - 1, \tag{A7}$$

$$\lambda_m = \begin{cases} \frac{2}{\Delta y^2}(\cos\frac{2\pi m}{N_y} - 1) & \text{DFT} \\ \frac{2}{\Delta y^2}(\cos\frac{\pi m}{N_y} - 1) & \text{DCT} \end{cases} \quad m = 0, \ldots, N_y - 1. \tag{A8}$$

For each mode, this is a tridiagonal system with $k = 0, \ldots, N_z + 1$, which is solved using Gaussian elimination. In uDALES, boundary conditions are not inserted into $\hat{f}_0$ and $\hat{f}_{N_z+1}$ in order to explicitly solve for the ghost cells $\hat{\pi}_0$ and $\hat{\pi}_{N_z+1}$. Instead, the equations for $k = 1$ and $k = N_z$ are modified in light of the top and bottom boundary conditions, which is numerically equivalent. Specifically, the bottom boundary condition is Neumann for all modes, i.e. $\hat{\pi}_{n,m,0} = \hat{\pi}_{n,m,1} \ \forall \ n, m$. This means that the $k = 1$ equation becomes

$$(a_1 + b_{n,m,1})\hat{\pi}_{n,m,1} + c_1 \hat{\pi}_{n,m,2} = \hat{f}_{n,m,1}. \tag{A9}$$

The top is Neumann for all modes other than $n = m = 0$ (the zero mode), in which case it is Dirichlet, i.e. $\hat{\pi}_{N_z+1} = -\hat{\pi}_{N_z}$. Applying the Neumann condition results in

$$a_{N_z-1}\hat{\pi}_{n,m,N_z-1} + (b_{n,m,N_z} + c_{N_z})\hat{\pi}_{n,m,N_z} = \hat{f}_{n,m,N_z}, \tag{A10}$$

and the Dirichlet condition for the zero mode results in

$$a_{N_z-1}\hat{\pi}_{0,0,N_z-1} + (b_{0,0,N_z} - c_{N_z})\hat{\pi}_{0,0,N_z} = \hat{f}_{0,0,N_z}. \tag{A11}$$

Numerically, this avoids the problem being ill-posed: since $b_{0,0,k} = 0$, the matrix describing the system of Eq. (A6) is singular if both the first and last rows are the same, which would be the case if both correspond to a Neumann condition. Setting a Dirichlet condition instead at the top is also a physically reasonable choice because the zero mode is equal to the plane-averaged pressure, which can be reasonably expected to be zero at the top of the domain. For inflow–outflow simulations, the pressure gradient at the top is accounted for in the boundary condition for vertical velocity $w \equiv u_z$. Given that the Dirichlet boundary condition for plane-averaged pressure is $\langle \pi \rangle_{N_z+1} = -\langle \pi \rangle_{N_z}$, the vertical velocity satisfies

$$\frac{w^*_{i,j,N_z+1} - w^n_{i,j,N_z+1}}{\Delta t} = \frac{2\langle p^n \rangle_{N_z}}{\Delta z_{N_z+1}} \quad \forall i, j, \tag{A12}$$

$$\frac{w^{n+1}_{i,j,N_z+1} - w^*_{i,j,N_z+1}}{\Delta t} = \frac{2\langle \pi^n \rangle_{N_z}}{\Delta z_{N_z+1}} \quad \forall i, j. \tag{A13}$$

## Appendix B: Reconstruction

If the distance to the facet $d$ is less than $z_0 e$, where $z_0$ is the momentum roughness length, the wall functions become

non-physical because there is a factor of $\log(d/z_0)$ (Suter et al., 2022). In these cases, it is necessary to use reconstruction. When using reconstruction, the flux is calculated at a point further from the facet in the facet normal direction, where the stencil should contain only fluid points. Once the flux has been calculated at this reconstruction point, it can be applied to the fluid boundary point because by definition the region in which the wall functions are valid is a constant flux layer. The location is found by projecting a ray in the normal direction from the fluid boundary point until it hits one of the edges of the cell. A common approach is to project a fixed distance into the adjacent cell, but this would mean it is possible that the surrounding points are at least two indices away, which is problematic if the flux point is on the edge of a decomposition pencil. This approach guarantees that all the points used in the stencil are known to the current rank.

Trilinear interpolation is used to determine the value of flow variables at the reconstruction point. To calculate a variable $\varphi$ at a point $(x, y, z)$ somewhere in cell $i, j, k$ of a given grid, i.e. $x_i \leq x \leq x_{i+1}$, $y_j \leq y \leq y_{j+1}$, and $z_k \leq z \leq z_{k+1}$, let $x_\mathrm{d} = (x - x_i)/(x_{i+1} - x_i) = (x - x_i)/\Delta x$, and similar for $y_\mathrm{d}$ and $z_\mathrm{d}$ CE5. Then the value is given by

$$
\begin{aligned}
\varphi(x, y, z) &= \varphi(x_i, y_j, z_k)(1 - x_\mathrm{d})(1 - y_\mathrm{d})(1 - z_\mathrm{d}) \\
&+ \varphi(x_{i+1}, y_j, z_k)x_\mathrm{d}(1 - y_\mathrm{d})(1 - z_\mathrm{d}) \\
&+ \varphi(x_i, y_{j+1}, z_k)(1 - x_\mathrm{d})y_\mathrm{d}(1 - z_\mathrm{d}) \\
&+ \varphi(x_{i+1}, y_{j+1}, z_k)x_\mathrm{d}y_\mathrm{d}(1 - z_\mathrm{d}) \\
&+ \varphi(x_i, y_j, z_{k+1})(1 - x_\mathrm{d})(1 - y_\mathrm{d})z_\mathrm{d} \\
&+ \varphi(x_{i+1}, y_j, z_{k+1})x_\mathrm{d}(1 - y_\mathrm{d})z_\mathrm{d} \\
&+ \varphi(x_i, y_{j+1}, z_{k+1})(1 - x_\mathrm{d})y_\mathrm{d}z_\mathrm{d} \\
&+ \varphi(x_{i+1}, y_{j+1}, z_{k+1})x_\mathrm{d}y_\mathrm{d}z_\mathrm{d}.
\end{aligned}
$$

Note that in this case since the reconstruction point is at one of the cell edges, the scheme for variables defined on this edge reduces to a bilinear interpolation. A 2D example for a scalar variable is shown in Fig. B1.

*Code and data availability.* The uDALES codebase and user manual is available on GitHub at https://github.com/uDALES/u-dales (last access: 4 August 2024) and on Zenodo at https://doi.org/10.5281/zenodo.10671714 (Grylls et al., 2024). The user manual includes guidance on the 2D domain decomposition and on running the pre-processing routines required to generate the necessary inputs for the immersed boundary method and the surface energy balance model. The dataset for the validation cases presented in this paper are available on Zenodo at https://doi.org/10.5281/zenodo.12510825 (Owens et al., 2024).

*Author contributions.* SOO carried out the bulk of the model development including the 2D domain decomposition (Sect. 3.1), the pressure solver (Sect. 3.2), and the geometry representation (Sect. 3.3) and was involved in the periodic neutral validation (Sect. 4.1) and the non-neutral validations (Sects. 4.3 and 4.4).

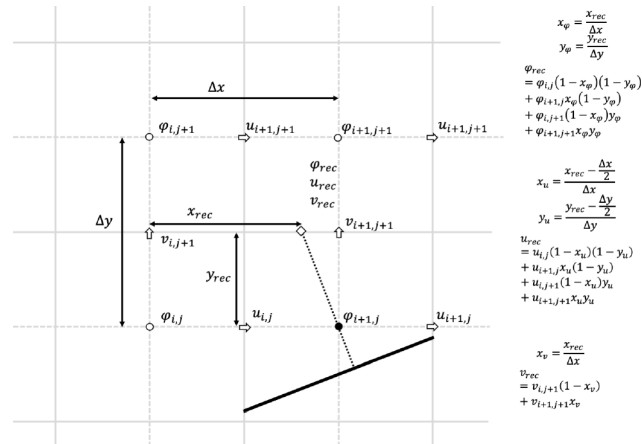

**Figure B1.** Reconstruction of flow variables in 2D at the point $\varphi_{i+1, j}$ using a bilinear interpolation scheme. A facet section is shown in solid black, the $c$-grid cell edges are solid grey, and the $u$- and $v$-grid cell edges are dotted grey (where they are staggered with respect to the $c$ grid). Note that the bilinear interpolation reduces to a linear one for the $v$ velocity.

DM contributed to model development, particularly with respect to pre-processing, and was involved in the periodic neutral validation (Sect. 4.1) and the indoor–outdoor validation (Sect. 4.2). CEW informed the model development and was involved with the surface energy balance validation (Sect. 4.4). PB supervised and contributed to the model development with respect to the use of the 2DECOMP&FFT library, including performing scaling tests (Sect. 3.1). MvR acquired funding, supervised the project as a whole, and conceived of the shortwave radiation algorithm (Sect. 3.4) and the shifting method for periodic simulations (Sect. 3.5). SOO prepared the paper with contributions from all co-authors.

*Competing interests.* The contact author has declared that none of the authors has any competing interests.

ther geographical representation in this paper. While Copernicus Publications makes every effort to include appropriate place names, the final responsibility lies with the authors.

*Financial support.* This work was funded under the embedded CSE programme of the ARCHER2 UK National Supercomputing Service (http://www.archer2.ac.uk, last access: 4 August 2024; project ARCHER2-eCSE05-3) and the Turbulence at the Exascale project (EP/W026686/1), which is part of the ExCALIBUR HPC programme funded by EPSRC.

*Review statement.* This paper was edited by Mohamed Salim and reviewed by five anonymous referees.

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

**Remarks from the language copy-editor**