# Peer review of "A conservative immersed boundary method for the multi-physics urban large-eddy simulation model uDALES v2.0"

_EGUsphere, 2024_

## Referee Comment (RC3)

**Review Report for Manuscript: A conservative immersed boundary method for the multi-physics urban large-eddy simulation model uDALES v2.0**

May 9, 2024

**1 General**

The manuscript titled "A conservative immersed boundary method for the multiphysics urban large-eddy simulation model uDALES v2.0" presents advancements in the uDALES framework, particularly focusing on the implementation of a conservative immersed boundary method (IBM) for urban surface representation. This method aims to address the challenges associated with complex urban geometries in microscale urban airflow simulations. While the paper demonstrates improvements over the previous version of uDALES and provides insights into the capabilities of the new method, several key issues need to be addressed before the manuscript can be considered for publication.

**2 Major Issues**

- **Radiation Interaction and Reflections**: One major issue concerns the clarity of the radiation interaction process and reflections in the manuscript. The explanation of how the new method of surface representation improves reflections is not adequately presented. While the authors briefly mention view factors, further details are needed to understand the specific improvements achieved with the new method. Additionally, the validation simulations do not adequately reflect the impact of the improved surface representation on radiation interactions. I believe further clarification and demonstration is needed.

- **Simplicity of Test Cases**: Another major concern is the simplicity of the test cases used for validation, which may not fully capture the complexity of urban environments. The validation simulations should consider a wider range of scenarios to assess the robustness and applicability of the new method in diverse urban settings. Incorporating more realistic urban configurations and environmental conditions would strengthen the validity and relevance of the findings. Otherwise, justification is missing here.

- **Representation of Resolved Vegetation**: The manuscript lacks sufficient detail regarding the representation of resolved vegetation in the simulations. Given the importance of vegetation in urban microclimate simulations, particularly in

influencing surface energy balances and pollutant dispersion, it is essential to provide a comprehensive description of how vegetation is incorporated into the model and validated in the simulations. Including relevant references and discussing the implications of vegetation representation would enhance the completeness of the manuscript.

- **Increased Computational Requirement**: The introduction of the new triangular surface representation may significantly increase the number of surfaces compared to traditional rectangular surfaces. The potential impact of this increase on the performance and computational requirements of the model should be clarified. Authors are encouraged to discuss any potential implications for computational efficiency and scalability.

- **Accuracy of Numerical Solution**: The scheme now resembles somewhat an unstructured grid, which may raise concerns about the accuracy of the numerical solution due to increased numerical errors. Authors should address whether this change affects the accuracy of the solution and discuss any measures taken to mitigate potential errors (if any). The discussion should compare the accuracy with the traditional unstructured gird given here: `https://doi.org/10.1016/j.buildenv.2008.11.010`.

**3  Minor Issues**

- **Introduction Citations**: The introduction section lacks citations to support the background information provided. For example, lines 28-29 and lines 35-36 may benefit from adding some references, such as `https://doi.org/10.1016/j.enbuild.2023.113324`, which exemplify the application of micorscale urban climate models in realistic urban environments. Also the radiation model of PALM in line 54 (`https://doi.org/10.5194/gmd-14-3095-2021` and `https://doi.org/10.5194/gmd-15-145-2022`)

- **Additional Models in Introduction**: Lines 37-39 should include references to other relevant models such as MITRAS, MISKAM, etc., to provide a comprehensive overview of existing approaches in the field.

- **Temperature Scale in Figure Legend**: The temperature scale in the legend of figure 1 has a minimum and maximum of 300K. Additionally, the caption of the figure should be more informative to provide clear context and interpretation of the results.

- **Relevance of Certain Text**: Lines 74 - 77 may not be necessary and could be considered for removal if they do not contribute to the manuscript.

- **Organization of Content**: Line 153 should be moved to the relevant subsection for better organization and clarity.

- **Confined with ARCHER2**: Lines 158-160 suggest a confinement to ARCHER2, which may limit the generalizability of the findings. Authors should clarify whether the method is applicable to other computing platforms and address any potential limitations in this regard.

- **Figure 14**: The presence of incoming longwave radiation in Figure 14 should be clearly indicated or discussed in the caption to provide a comprehensive understanding of the depicted variables.

**4    General Opinion**

Overall, the manuscript presents valuable contributions to the field of urban microscale airflow simulation, particularly through the implementation of the conservative immersed boundary method in uDALES v2.0. Despite the major issues identified, the study demonstrates promising advancements in addressing the challenges associated with complex urban geometries. Addressing the identified issues, particularly regarding radiation interactions, test case complexity, and vegetation representation, will further strengthen the manuscript and enhance its impact in the field.

---

## Author Comment (AC4)

**Response to Referee #6:**

**General Comments**

In the present study, the authors have taken advantage of the existing knowledge to produce a new version (v2.0) of the uDALES code, demonstrating in addition the positive impact of the changes on prediction capabilities.

The effort here is to present the improvements on uDALES 1.0 to become uDALES 2.0. Therefore common descriptions are better to be limited and give more emphasis in the differences.

It is assumed that each application selected serves a specific objective. It is good this specific objective to be described and at the end of the specific exercise, conclusions need to be drawn in relation to this objective. A generic objective is to compare results of uDALES v2.0 and uDALES v1.0 and commenting on the differences and their causes. Please make sure that those principles are kept to a large degree, for all applications presented.

It is most probable that some limitations still persist in the v2.0 version. It would be helpful, in a separate chapter, such limitations are discussed including the way forward.

**Response:** We thank the reviewer for these comments. In the revised manuscript, we have emphasised and expanded on differences between uDALES v1 and v2 in Sect. 3. We have also provided a more thorough conclusion of each of the validation cases. Finally, we have included a discussion of model limitations, though we thought it was more appropriate as the the penultimate paragraph in the Conclusion rather than a separate chapter.

**Specific Comments**

**Comment 1:** Lines 41- 43 "RANS only resolves the mean flow, and entirely relies on turbulence modelling to incorporate the effect of fluctuating components. As a result, RANS often fails to accurately reproduce transient flow features and quantities such as turbulent kinetic energy (van Hooff et al., 2017)." . It is not evident that the RANS characteristics prevent them to reproduce transient flow features. Do you mean unsteady flow features ? . Concerning turbulent kinetic energy predictions with RANS Models probab;y there are models that claim they can do this. See for example : Bartzis, J,G , Boundary-Layer Meteorology (2005) 116: 445–459 ,DOI 10.1007/s10546-004-7404-y

**Response 1:** By the phrase 'transient flow features' in lines 41-43, we indeed meant the 'unsteady flow features' i.e. the time dependent evolution of the flow. While simulating the cross-ventilation for a isolated building, van Hooff et al. (2017) showed that all RANS models, namely shear stress transport (SST), renormalization group (RNG) or Reynolds stress model (RSM), failed in reproducing the turbulent kinetic energy because steady RANS could not capture the vertical oscillation of the jet entering through the windward window of the building. We agree with the reviewer that the original statement was too broad. We have rephrased the sentence in the revised manuscript as "... these models often fail to reproduce unsteady flow features, leading to inaccurate prediction of quantities like turbulent kinetic energy for the flows that involve time dependent modulation (Moonen et al., 2012; van Hooff et al., 2017; Blocken, 2018; Vita et al., 2020)".

**Comment 2:** Lines 45 -46 : "... LES is inherently more accurate than RANS...." . It would be better to say "... LES is expected to be more accurate than RANS". We should not forget that we have to apply LES correctly and this not always easy for open flows such as the atmospheric boundary layer.

**Response 2:** We agree, and revised the sentence as ".. LES is expected to be more accurate than RANS ... provided that the boundary conditions, including the surface, are modelled accurately, which is challenging for urban boundary layer flows."

**Comment 3:** Lines 105-107 : Concerning horizontal grid, it seems that there is a different treatment for x- and y -directions. In real problems this might impose limited degree of freedom in selecting x- and y- directions. In other words is this selection application dependent ?

**Response 3:** In practice, it is true that the $x$- and $y$- directions almost always conceptually correspond to the spanwise and streamwise directions respectively. However, this does not have to be the case when running periodic simulations, and when using inflow-outflow boundaries with fixed values. When using a time-varying inflow condition though, this can only be specified on the $x$ inlet plane. We have included this discussion in the conclusion.

**Comment 4:** Lines 110 -113 ; The inlet flow conditions are briefly discussed. The key problem here is how the upwind large eddies are introduced into the computation domain, keeping in mind that any introduced error persists downwind. Please expand.

**Response 4:** We have added (on Line 111 onwards): "The resolution in the $y$- and $z$-directions of the main simulation necessarily equals that of the precursor, meaning there no interpolation is required in these directions, and the flow variables on the output plane of the precursor can be copied directly to the inlet of the main simulation. Technically, the Dirichlet inflow boundary condition for the main simulation is enforced at $x = 0$, so $u$ can be set directly as it defined on cell edges. The other variables are (linearly) interpolated in the $x$-direction as they are defined on cell centres."

Alternatively, the reviewer may be referencing the need for a 'run-up' region before the flow reaches the geometry. We have also added to the end of Line 113: " It is good practice to have a reasonable distance between the boundaries and the object of interest, in order to allow the flow to adjust (Tominaga et al., 2008)."

**References**

Blocken, B.: LES over RANS in building simulation for outdoor and indoor applications: A foregone conclusion?, in: Building Simulation, vol. 11, pp. 821–870, Springer, 2018.

Moonen, P., Dorer, V., and Carmeliet, J.: Effect of flow unsteadiness on the mean wind flow pattern in an idealized urban environment, Journal of wind engineering and industrial aerodynamics, 104, 389–396, 2012.

Tominaga, Y., Mochida, A., Yoshie, R., Kataoka, H., Nozu, T., Yoshikawa, M., and Shirasawa, T.: AIJ guidelines for practical applications of CFD to pedestrian wind environment around buildings, Journal of wind engineering and industrial aerodynamics, 96, 1749–1761, 2008.

van Hooff, T., Blocken, B., and Tominaga, Y.: On the accuracy of CFD simulations of cross-ventilation flows for a generic isolated building: Comparison of RANS, LES and experiments, Building and Environment, 114, 148–165, 2017.

Vita, G., Salvadori, S., Misul, D. A., and Hemida, H.: Effects of inflow condition on RANS and LES predictions of the flow around a high-rise building, Fluids, 5, 233, 2020.

---

## Author Comment (AC5)

**Response to Referee #3:**

**General Overview:** The manuscript titled "A conservative immersed boundary method for the multi-physics urban large-eddy simulation model uDALES v2.0" presents advancements in the uDALES framework, particularly focusing on the implementation of a conservative immersed boundary method (IBM) for urban surface representation. This method aims to address the challenges associated with complex urban geometries in microscale urban air flow simulations. While the paper demonstrates improvements over the previous version of uDALES and provides insights into the capabilities of the new method, several key issues need to be addressed before the manuscript can be considered for publication.

**Response:** We thank the reviewer for this thorough review. Below, we present a point-by-point response to the issues. Given that addressing the major issues in the manuscript required a fairly large number of changes in multiple locations, for clarity we generally do not include the exact changes made here. Instead, we describe where and how the manuscript has been revised.

**Major Issues:**

**Radiation Interaction and Reflections:** One major issue concerns the clarity of the radiation interaction process and reflections in the manuscript. The explanation of how the new method of surface representation improves reflections is not adequately presented. While the authors briefly mention view factors, further details are needed to understand the specific improvements achieved with the new method. Additionally, the validation simulations do not adequately reflect the impact of the improved surface representation on radiation interactions. I believe further clarification and demonstration is needed.

**Response:** We did not go into detail in this paper since the handling of reflected radiation is conceptually unchanged from uDALES v1. One difference is that the view factor calculation is done using the open-source code View3D, which is capable of handling complex triangular meshes. This however does not change the reflection calculations. The algorithm used to account for reflections is an iterative method that continues until all radiation up to a specified threshold is either absorbed or is reflected to the sky (Suter et al., 2022).

Note that we do not claim that the new geometry representation improves reflections per se; the radiation calculation will be more accurate (for non-grid-aligned geometries) because the walls are no longer limited to being either parallel or perpendicular. The reason the reflections are important is shown in Sect. 4.4 - if the geometry is approximated as being aligned to the grid by using lots of small facets, there can be reflections between facets on the same wall that clearly would not occur otherwise. Such problems would only be compounded if using a geometry with reflections between different walls, for example a street canyon (such a case was in fact considered for Sect. 4.4). Since the reflection algorithm has not changed, we feel that the case presented in the paper validates the new geometry specification sufficiently. To address the reviewer's concerns, we have expanded Sect. 3.4 and emphasised what has changed from v1 and v2, and how exactly this impacts performance and accuracy.

**Simplicity of Test Cases:** Another major concern is the simplicity of the test cases used for validation, which may not fully capture the complexity of urban environments. The validation simulations should consider a wider range of scenarios to assess the robustness and applicability of the new method in diverse urban settings. Incorporating more realistic urban configurations and environmental conditions would strengthen the validity and relevance of the findings. Otherwise, justification is missing here.

**Response:** We acknowledge that the cases presented do not capture the full complexity of urban environments. The intention in this paper is to describe and validate parts of the model that have been changed from v1 to v2, and explain the differences

in the results, and the test cases are therefore simple by design. In particular, the paper introduces (to the authors' knowledge) a novel approach for urban LES on Cartesian grids - namely the use of an IBM capable of handling non-grid-aligned surfaces that includes heat transfer. The priority is therefore to demonstrate the validity of this approach at a fundamental level, as bottom-up validation is standard practice when making changes of this magnitude. Using more complex cases (that are not focused on isolating a particular aspect of the model) would make explaining the differences more challenging, and potentially even mask conceptual and implementation errors.

Having made the argument that simple cases are appropriate given the aims of this paper, we acknowledge the need for urban models, including ours, to be validated and compared to each other for realistic urban settings. We have had preliminary discussions to perform an inter-comparison exercise with different urban codes developed by other groups, but this is well beyond the scope of the current paper.

**Representation of Resolved Vegetation:** The manuscript lacks sufficient detail regarding the representation of resolved vegetation in the simulations. Given the importance of vegetation in urban microclimate simulations, particularly in influencing surface energy balances and pollutant dispersion, it is essential to provide a comprehensive description of how vegetation is incorporated into the model and validated in the simulations. Including relevant references and discussing the implications of vegetation representation would enhance the completeness of the manuscript.

**Response:** A complete description of how resolved vegetation (trees) are modelled in uDALES is available in Grylls and van Reeuwijk (2021); Suter et al. (2022). This is not an aspect of the model that has changed from v1 and v2, so we prefer not to repeat this description in detail or to re-validate this part of the model. We have further emphasised this in Sect. 3.3.

**Increased Computational Requirement:** The introduction of the new triangular surface representation may significantly increase the number of surfaces compared to traditional rectangular surfaces. The potential impact of this increase on the performance and computational requirements of the model should be clarified. Authors are encouraged to discuss any potential implications for computational efficiency and scalability.

**Response:** Note that the computational performance of the IBM scales linearly with the total number of facet *sections*, rather than the number of facets, and the number of sections depends on total surface area and the grid resolution - we have added this point to Sect. 3.3.2. However, the shortwave radiation-related routines do scale with the number of facets ($N$), specifically the view factor calculation is $\mathcal{O}(N^2)$, and direct solar radiation calculation was approximately $\mathcal{O}(N^2)$ in uDALES v1 and is now $\mathcal{O}(N)$ in v2. Note that the calculation of net shortwave radiation is a pre-processing step, and so does not affect the performance of the code in the sense discussed in Sect. 3.1.

With respect to triangles vs quadrilaterals, indeed converting a given quad mesh to a tri mesh by repeatedly dividing each quad by 2 will result in at least twice as many faces – see Sect. 4.1, and also 4.3/4.4 where uDALES v2 uses more facets than v1 in the non-rotated case. However, it is not necessary to generate the mesh in this way - see Sect. 4.2, where it is not clear that fewer quads than tris can be used to represent the building. Also, the fact that the radiation routines scale better and practically run faster means that v2 will tend to perform better even if $N$ is larger. We have added a discussion of these points to Sects. 3.3 and 3.4.

**Accuracy of Numerical Solution:** The scheme now resembles somewhat an unstructured grid, which may raise concerns about the accuracy of the numerical solution due to increased numerical errors. Authors should address whether this change

affects the accuracy of the solution and discuss any measures taken to mitigate potential errors (if any). The discussion should compare the accuracy with the traditional unstructured gird given here: https://doi.org/10.1016/j.buildenv.2008.11.010.

75 **Response:** While the geometry representation is unstructured, uDALES v2 still uses a Cartesian computational grid, with an immersed boundary method to model the geometry. The governing equations and discretisation of the LES are unchanged from v1, therefore the accuracy is too. The key difference is how the surface fluxes are calculated (using the wall functions). For non-grid-aligned surfaces, there are some considerations to do with grid resolution (relating to the distance to the wall) that were not included in the original text and have been added in Sect. 3.3.2 and Appendix B. This issue aside, the wall functions

80 should not be affected much by spatial discretisation (at least not more so in v2 than v1) as they are physical parametrisations. We have added a mention of the use of unstructured meshes, using the provided reference, in the Introduction.

**Minor Issues:**

**Introduction Citations:** The introduction section lacks citations to support the background information provided. For example, lines 28-29 and lines 35-36 may benefit from adding some references, such as https://doi.org/10.1016/j.enbuild.2023.113324,

85 which exemplify the application of micorscale urban climate models in realistic urban environments. Also the radiation model of PALM in line 54 (https://doi.org/10.5194/gmd-14-3095-2021 and https://doi.org/10.5194/gmd-15-145-2022)
**Response:** We have added the references to PALM in the Introduction, and added the other reference to the Conclusion as an example of the kind of studies that are now possible with uDALES v2.0.

**Additional Models in Introduction:** Lines 37-39 should include references to other relevant models such as MITRAS,

90 MISKAM, etc., to provide a comprehensive overview of existing approaches in the field.
**Response:** We have added a reference to MITRAS (Salim et al., 2018) in the Introduction.

**Temperature Scale in Figure Legend:** The temperature scale in the legend of figure 1 has a minimum and maximum of 300K. Additionally, the caption of the figure should be more informative to provide clear context and interpretation of the results.
**Response:** The temperature field is not in fact relevant. We therefore have opted to replace Fig. 1 (see below), which shows

95 the problem with representing realistic geometries in uDALES v1 with respect to radiation.

**Relevance of Certain Text:** Lines 74 - 77 may not be necessary and could be considered for removal if they do not contribute to the manuscript.
**Response:** We would prefer to keep these lines, as some readers might find this useful.

**Organization of Content:** Line 153 should be moved to the relevant subsection for better organization and clarity.
100 **Response:** We have moved this sentence to the Introduction (Ln 73).

**Confined with ARCHER2:** Lines 158-160 suggest a confinement to ARCHER2, which may limit the generalizability of the findings. Authors should clarify whether the method is applicable to other computing platforms and address any potential limitations in this regard.
**Response:** As described in the text, the parallelisation of uDALES is based on 2DECOMP&FFT, which has been used as

105 the basis for several open source projects, most notably the open-source DNS framework XCompact3D (Bartholomew et al., 2020), and has been used across all major supercomputing (CPU-based) platforms worldwide. We are currently working on

[Figure]

$K^*$ [W m$^{-2}$]

**Figure 1.** Net shortwave radiation ($K^*$) on buildings for the demonstration case in Suter et al. (2022). Note that the ground surfaces have been excluded from the visualisation for clarity.

preparing the code for use with GPUs. At the end of Sect. 3.1 we have added: "2DECOMP&FFT has been developed for cross-platform usage and has been tested on all major supercomputer architectures (Li and Laizet, 2010; Rolfo et al., 2023), ensuring excellent portability of uDALES.".

110 **Figure 14:** The presence of incoming longwave radiation in Figure 14 should be clearly indicated or discussed in the caption to provide a comprehensive understanding of the depicted variables.

**Response:** We have changed Ln 496 onwards to: "... the leeward and side faces have reduced incoming longwave radiation for the same reasons given for the shortwave. This is consistent with the fact that the net longwave in Fig. 14 is less negative, and the outgoing longwave is directly determined by surface temperature, which is lower in the v1 rotated case. The temperature
115 of the leeward face is lower because it has less available energy due to the reduced shortwave, which also acts to reduce the sensible heat and conductive fluxes...".

**References**

Bartholomew, P., Deskos, G., Frantz, R. A., Schuch, F. N., Lamballais, E., and Laizet, S.: Xcompact3D: An open-source framework for solving turbulence problems on a Cartesian mesh, SoftwareX, 12, 100 550, 2020.

120 Grylls, T. and van Reeuwijk, M.: Tree model with drag, transpiration, shading and deposition: Identification of cooling regimes and large-eddy simulation, Agricultural and Forest Meteorology, 298, 108 288, 2021.

Li, N. and Laizet, S.: 2decomp & fft-a highly scalable 2d decomposition library and fft interface, in: Cray user group 2010 conference, pp. 1–13, 2010.

Rolfo, S., Flageul, C., Bartholomew, P., Spiga, F., and Laizet, S.: The 2DECOMP&FFT library: an update with new CPU/GPU capabilities,
125 Journal of Open Source Software, 8, 5813, 2023.

Salim, M. H., Schlünzen, K. H., Grawe, D., Boettcher, M., Gierisch, A. M., and Fock, B. H.: The microscale obstacle-resolving meteorological model MITRAS v2. 0: model theory, Geoscientific Model Development, 11, 3427–3445, 2018.

Suter, I., Grylls, T., Sützl, B. S., Owens, S. O., Wilson, C. E., and van Reeuwijk, M.: uDALES 1.0: a large-eddy simulation model for urban environments, Geoscientific Model Development, 15, 5309–5335, 2022.

---

## Author Response (AR1)

**Author's response**

We thank the editor for their time in reviewing our manuscript. We feel we have addressed all the referee comments, and here we present our point-by-point response to them, as well as any other changes made to the manuscript. Referee comments are shown in italic type, followed by a description of any changes made in roman type. Where the line numbers are given in our responses, this refers to the revised manuscript.

**Referee Comments #1 & #4:**

*This paper presents enough materials to describe the upgraded key features from uDALES v1.0 to uDALES v2.0 (LES), as well as the crucial technique details. Four cases were tested and validated against the data in the literature, showing the success of the upgrading. I am very pleased to see that the authors have published the uDALES v2.0 codebase and user manual for open access. The MS is concise and is well prepared. I would recommend 'accept' for publication, subject to addressing some minor comments below:*

- *Lines 8-9. Please rephrase this sentence 'Good agreement . . . '*

  Changed to 'We observe good agreement...'

- *Eq. (2), in the convection term $u_i$ to change to $u_j$. Give a bit more details about $K_\mathrm{h}$ (it should be SGS eddy diffusivity).*

  We have not changed Eq. (2), as we prefer to use the usual form for advection, i.e. the first velocity component ($u_j$) is advecting and the second is being advected ($u_i$). Also, we have added more detail on how $K_\mathrm{h}$ is calculated in the sentence following Eq. (2).

- *Line 112, change 'transported' to 'convected'.*

  Changed (on Line 122) to 'advected' as 'convected' has unwanted connotations of vertical motion/heat transfer.

- *Eq. (5), define RH, $q_\mathrm{sat}$.*

  No change - the reviewer later said they "*missed the text above the equation 5*".

- *Lines 142-144. "Note that the surface energy balance is generally evolved using a larger timestep than the LES, therefore these fluxes are time averaged". This is not clear enough. These fluxes are constant during the LES, or vary slowly in a quasi-steady manner?*

  Changed Lines 162-164 to: "Note that the turbulent fluxes ($H$ and $E$) are calculated at every LES timestep, but the surface energy balance is generally evolved using a larger timestep than the LES. The fluxes used to solve for the facet temperatures are therefore the time-average of the fluxes at LES timesteps."

- *Section 3 title 'Improvements' to change to 'Upgraded key features from uDALES v1.0 to v2.0'*

  Changed to 'Upgraded features', as we thought the reviewer's suggestion too verbose.

- *Sect. 3.2. As in the inlet-outlet direction (streamwise), only the Neumann BC can be used. Therefore, the constant pressure boundary condition cannot be used. Given all boundary conditions for pressure are Neumann, any specific treatment is given to settle down a unique solution?*

No change - this information is given in Appendix A. In summary: to make the Poisson equation for pressure well-posed, a Dirichlet boundary condition is used for the plane-averaged pressure at the top.

- *Line 228. Non-slip boundary condition means non-zero shear, which is wall-normal momentum flux. Please comment.*

  We do initially impose a free-slip (zero flux) boundary condition by negating any diffusion, and then add shear stress according to the wall function (thus making it no slip). We have changed Lines 249-253 to "uDALES assumes the geometry is non-porous and stationary, which are modelled using no-slip and no-penetration boundary conditions. The fluxes of momentum and scalars across boundaries are prescribed by wall functions, rather than being determined by the advection and diffusion terms in the governing equations (Eq. (2) and Eq. (3)). This condition, *i.e.* zero flux other than what is prescribed by the wall functions, is particularly important for scalars."

- *Lines 245-246. Please describe a bit more of the interpolation here, is it a 2nd-order accuracy linear interpolation?*

  No change - we don't actually carry out this interpolation, so further detail is not necessary.

- *Lines 335-338. The 'real' turbulence generated by the driver simulations using periodic lateral boundary conditions is governed by the upper and lower boundaries, which are just a simplified condition (but not 100% 'real'). Please comment. I suggest [changing to:] "They typically use periodic inlet-outlet boundary conditions, and their advantage is that the technique is simpler to implement requiring less prescribed information, as opposed to the more sophisticated methods, such as the synthetic turbulence generation (e.g. Xie and Castro, 2008)."*

  Changed Lines 384-390 to: "They typically use periodic boundary conditions, and their advantage is that all spatial and temporal statistics of the eddies (e.g. two-point correlations) are representative of real turbulence provided the precursor simulation has an appropriate aspect ratio domain. This is in contrast to the common alternative approach of quantities being prescribed (on the inlet) and turbulence being generated synthetically (*e.g.* Xie and Castro, 2008). The precursor technique can be simpler to implement as it requires less prior information, though given the flow is strongly determined by the bottom boundary condition, trial and error with respect to the geometry is often required to obtain the desired turbulence statistics (if trying to match specific data)."

- *Line 374. 'These differences are most likely caused by differences in the inflow conditions above $z/h_m = 5$ (Fig. 8c)'. The upper boundary conditions in the LES might be one more reason to cause the discrepancy? Please comment.*

  Changed Lines 426-428 to "These differences are most likely for the same reasons discussed in Xie et. al. (2008), namely that the experimental and numerical studies each use a different domain height."

- *Section 4.2. Please give the dimensions of the LES domain in the text, although they are shown in Fig. 9.*

  Added on Line 441.

- *Section 4.3. The Ri number is kept the same as in Richards et al (2006) and Boppana et al (2013), whereas the Reynolds number is increased by 100 times (or kept the same as Richards et al (2006))? If not, please comment on this.*

  Added on Lines 483-488: "The Reynolds number ($Re$) for urban flows is typically sufficiently large for the viscous stresses to be negligible compared to the turbulent stresses and pressure, i.e. the flow is essentially independent of $Re$. This means the geometry scale can be changed while keeping flow velocity constant (and thus changing $Re$) and obtain

similar results, as in Sect. 4.1. However, $Ri$ is a measure of the relative strength of buoyancy to the mean kinetic energy, and this ratio must stay the same otherwise the character of the flow will change."

- *Section 4.4. Please give the Ri number in the current problem, with a discussion on the capability of uDALES for larger Ri flows. To be consistent with Sect. 4.3, use the facet-average temperature to estimate the Ri number.*

  Added on Lines 551-552: "The temperature difference between the surface of the leeward face and $\theta_{\mathrm{ref}}$ is approximately 16 times higher than in Sect. 4.3, thus yielding $Ri \approx -10$. This indicates that buoyancy is more significant for this case."

**Referee Comment #2:**

*I found the presentation to be clear, and the topic is definitely appropriate. I am not qualified to review the technical aspects of the manuscript.*

No changes were made in response to this comment.

**Referee Comment #3:**

***General Overview:*** *The manuscript titled "A conservative immersed boundary method for the multi-physics urban large-eddy simulation model uDALES v2.0" presents advancements in the uDALES framework, particularly focusing on the implementation of a conservative immersed boundary method (IBM) for urban surface representation. This method aims to address the challenges associated with complex urban geometries in microscale urban air flow simulations. While the paper demonstrates improvements over the previous version of uDALES and provides insights into the capabilities of the new method, several key issues need to be addressed before the manuscript can be considered for publication.*

***Major Issues:***

- ***Radiation Interaction and Reflections:*** *One major issue concerns the clarity of the radiation interaction process and reflections in the manuscript. The explanation of how the new method of surface representation improves reflections is not adequately presented. While the authors briefly mention view factors, further details are needed to understand the specific improvements achieved with the new method. Additionally, the validation simulations do not adequately reflect the impact of the improved surface representation on radiation interactions. I believe further clarification and demonstration is needed.*

  We did not go into detail in this paper since the handling of reflected radiation is conceptually unchanged from uDALES v1. One difference is that the view factor calculation is done using the open-source code View3D, which is capable of handling complex triangular meshes. This however does not change the reflection calculations. The algorithm used to account for reflections is an iterative method that continues until all radiation up to a specified threshold is either absorbed or is reflected to the sky (Suter et al., 2022).

  Note that we do not claim that the new geometry representation improves reflections per se; the radiation calculation will be more accurate (for non-grid-aligned geometries) because the walls are no longer limited to being either parallel or perpendicular. The reason the reflections are important is shown in Sect. 4.4 - if the geometry is approximated as being aligned to the grid by using lots of small facets, there can be reflections between facets on the same wall that clearly would not occur otherwise. Such problems would only be compounded if using a geometry with reflections between different

walls, for example a street canyon (such a case was in fact considered for Sect. 4.4). Since the reflection algorithm has not changed, we feel that the case presented in the paper validates the new geometry specification sufficiently. To address the reviewer's concerns, we have expanded Sect. 3.4, in particular Lines 367-375, and emphasised what has changed from v1 and v2, and how exactly this impacts performance and accuracy.

– *Simplicity of Test Cases: Another major concern is the simplicity of the test cases used for validation, which may not fully capture the complexity of urban environments. The validation simulations should consider a wider range of scenarios to assess the robustness and applicability of the new method in diverse urban settings. Incorporating more realistic urban configurations and environmental conditions would strengthen the validity and relevance of the findings. Otherwise, justification is missing here.*

We acknowledge that the cases presented do not capture the full complexity of urban environments. The intention in this paper is to describe and validate parts of the model that have been changed from v1 to v2, and explain the differences in the results, and the test cases are therefore simple by design. In particular, the paper introduces (to the authors' knowledge) a novel approach for urban LES on Cartesian grids - namely the use of an IBM capable of handling non-grid-aligned surfaces that includes heat transfer. The priority is therefore to demonstrate the validity of this approach at a fundamental level, as bottom-up validation is standard practice when making changes of this magnitude. Using more complex cases (that are not focused on isolating a particular aspect of the model) would make explaining the differences more challenging, and potentially even mask conceptual and implementation errors.

Having made the argument that simple cases are appropriate given the aims of this paper, we acknowledge the need for urban models, including ours, to be validated and compared to each other for realistic urban settings. We have had preliminary discussions to perform an inter-comparison exercise with different urban codes developed by other groups, but this is well beyond the scope of the current paper. We have added a discussion of these points to the Conclusion (Lines 601-605).

– *Representation of Resolved Vegetation: The manuscript lacks sufficient detail regarding the representation of resolved vegetation in the simulations. Given the importance of vegetation in urban microclimate simulations, particularly in influencing surface energy balances and pollutant dispersion, it is essential to provide a comprehensive description of how vegetation is incorporated into the model and validated in the simulations. Including relevant references and discussing the implications of vegetation representation would enhance the completeness of the manuscript.*

A complete description of how resolved vegetation (trees) are modelled in uDALES is available in Grylls and van Reeuwijk (2021); Suter et al. (2022). This is not an aspect of the model that has changed from v1 and v2, so we prefer not to repeat this description in detail or to re-validate this part of the model. We have emphasised this in Sect. 3.3 on Lines 240-241.

– *Increased Computational Requirement: The introduction of the new triangular surface representation may significantly increase the number of surfaces compared to traditional rectangular surfaces. The potential impact of this increase on the performance and computational requirements of the model should be clarified. Authors are encouraged to discuss any potential implications for computational efficiency and scalability.*

Note that the computational performance of the IBM scales linearly with the total number of facet *sections*, rather than the number of facets, and the number of sections depends on total surface area and the grid resolution - we have added

this point to Sect. 3.3.2 (Lines 327-328). However, the shortwave radiation-related routines do scale with the number of facets ($N$), specifically the view factor calculation is $\mathcal{O}(N^2)$, and direct solar radiation calculation was approximately $\mathcal{O}(N^2)$ in uDALES v1 and is now $\mathcal{O}(N)$ in v2. We have also added on Lines 380-381 "note that the shortwave radiation calculation occurs as a pre-processing step, meaning it does not affect the performance of simulations in the sense discussed in Sect. 3.1"

With respect to triangles vs quadrilaterals, indeed converting a given quad mesh to a tri mesh by repeatedly dividing each quad by 2 will result in at least twice as many faces – see Sect. 4.1, and also 4.3/4.4 where uDALES v2 uses more facets than v1 in the non-rotated case. However, it is not necessary to generate the mesh in this way - see Sect. 4.2, where it is not clear that fewer quads than tris can be used to represent the building. Also, the fact that the radiation routines scale better and practically run faster means that v2 will tend to perform better even if $N$ is larger. We have added a discussion of these points to Sects. 3.3 (Lines 242-245) and 3.4 (Lines 379-380).

– **_Accuracy of Numerical Solution:_** _The scheme now resembles somewhat an unstructured grid, which may raise concerns about the accuracy of the numerical solution due to increased numerical errors. Authors should address whether this change affects the accuracy of the solution and discuss any measures taken to mitigate potential errors (if any). The discussion should compare the accuracy with the traditional unstructured grid given here: Hefny and Ooka (2009)._

While the geometry representation is unstructured, uDALES v2 still uses a Cartesian computational grid, with an immersed boundary method to model the geometry. The governing equations and discretisation of the LES are unchanged from v1, therefore the accuracy is too. The key difference is how the surface fluxes are calculated (using the wall functions). For non-grid-aligned surfaces, there are some considerations to do with grid resolution (relating to the distance to the wall) that were not included in the original text and have been added in Sect. 3.3.2 (Lines 257-261, 303-307 and 329-331) and Appendix B (Lines 670-671). This issue aside, the wall functions should not be affected much by spatial discretisation (at least not more so in v2 than v1) as they are physical parametrisations. We have added a mention of the use of unstructured meshes, using the provided reference, in the Introduction (Line 52).

**Minor Issues:**

– **_Introduction Citations:_** _The introduction section lacks citations to support the background information provided. For example, lines 28-29 and lines 35-36 may benefit from adding some references, such as Anders et al. (2023), which exemplify the application of microscale urban climate models in realistic urban environments. Also the radiation model of PALM in line 54 (Krč et al., 2021; Salim et al., 2022)._

We have added the references to PALM in the Introduction (Lines 58 and 64), and added the other reference to the Conclusion (Line 594) as an example of the kind of studies that are now possible with uDALES v2.0.

– **_Additional Models in Introduction:_** _Lines 37-39 should include references to other relevant models such as MITRAS, MISKAM, etc., to provide a comprehensive overview of existing approaches in the field._

We have added a reference to MITRAS (Salim et al., 2018) in the Introduction (Line 37).

- **Temperature Scale in Figure Legend:** *The temperature scale in the legend of figure 1 has a minimum and maximum of 300K. Additionally, the caption of the figure should be more informative to provide clear context and interpretation of the results.*

The temperature field is not in fact relevant. We therefore have opted to replace Fig. 1, which now shows the problem with representing realistic geometries in uDALES v1 with respect to radiation.

- **Relevance of Certain Text:** *Lines 74 - 77 may not be necessary and could be considered for removal if they do not contribute to the manuscript.*

We would prefer to keep these lines (81-84), as some readers might find this useful.

- **Organization of Content:** *Line 153 should be moved to the relevant subsection for better organization and clarity.*

We have moved this sentence to the Introduction (Line 80).

- **Confined with ARCHER2:** *Lines 158-160 suggest a confinement to ARCHER2, which may limit the generalizability of the findings. Authors should clarify whether the method is applicable to other computing platforms and address any potential limitations in this regard.*

As described in the text, the parallelisation of uDALES is based on 2DECOMP&FFT, which has been used as the basis for several open source projects, most notably the open-source DNS framework XCompact3D (Bartholomew et al., 2020), and has been used across all major supercomputing (CPU-based) platforms worldwide. We are currently working on preparing the code for use with GPUs. At the end of Sect. 3.1 (Lines 201-203) we have added: "2DECOMP&FFT has been developed for cross-platform usage and has been tested on all major supercomputer architectures (Li and Laizet, 2010; Rolfo et al., 2023), ensuring excellent portability of uDALES."

- **Figure 14:** *The presence of incoming longwave radiation in Figure 14 should be clearly indicated or discussed in the caption to provide a comprehensive understanding of the depicted variables.*

We have added on Lines 568-573: "... the leeward and side faces have reduced incoming longwave radiation for the same reasons given for the shortwave. This is consistent with the fact that the net longwave in Fig. 14 is less negative, and the outgoing longwave is directly determined by surface temperature, which is lower in the v1 rotated case. The temperature of the leeward face is lower because it has less available energy due to the reduced shortwave, which also acts to reduce the sensible heat and conductive fluxes...".

**Referee Comment #5:**

*This study shows result from uDALES v2.0. model which is an upgraded version uDALES v1.0, and compared results with the previous version of uDALES v2.0. and existing literatures. The upgraded version (i.e., uDALES v2.0.) results are consistent with the previous version (i.e., uDALES v1.0) and literatures, and have flexibility to study larger urban areas and run longer simulations. I found this manuscript well written and explain, although, I am not much expert on the technical aspects of the manuscript. I believe this manuscript is suitable for publication in GMD but I have few minor comments that I think would improve the manuscript:*

205     – *I noticed in the tile that author refers to uDALES v2.0. as a model, whereas in the abstract and rest of the manuscript, author refers to uDALES v2.0 as a framework, which is a bit confusing to me. Therefore, I would suggest stick with one of them to be consistent.*

We have changed 'framework' to 'model' or 'modelling tool' throughout the revised manuscript.

– *In line 13, it is not clear to me what trend author is referring there.*

210     We have removed 'If current trends continue' on Ln 13, as it is not in fact necessary to the point being made.

– *Line 36-39, I would suggest splitting the sentence, as it is a very long one to follow what information author wants to convey.*

We have simplified Lines 36-38.

**Referee Comment** #6:

215     *General Comments*

– *In the present study , the authors have taken advantage of the existing knowledge to produce a new version (v2.0) of the uDALES code, demonstrating in addition the positive impact of the changes on prediction capabilities. The effort here is to present the improvements on uDALES 1.0 to become uDALES 2.0. Therefore common descriptions are better to be limited and give more emphasis in the differences.*

220     We have emphasised and expanded on differences between uDALES v1 and v2 in Sect. 3, particularly Sect. 3.4.

– *It is assumed that each application selected serves a specific objective. It is good this specific objective to be described and at the end of the specific exercise, conclusions need to be drawn in relation to this objective. A generic objective is to compare results of uDALES v2.0 and uDALES v1.0 and commenting on the differences and their causes. Please make sure that those principles are kept to a large degree, for all applications presented.*

225     We have added a sentence to each of the validation cases' introductions to better motivate them (Lines 412-413, 435-436, 469-472, and 525-527), as well as to their conclusions (Lines 428-430, 459-461, 518-522, and 574-578)

– *It is most probable that some limitations still persist in the v2.0 version. It would be helpful, in a separate chapter, such limitations are discussed including the way forward.*

We have included a discussion of model limitations, though we thought it was more appropriate as the the penultimate
230     paragraph in the Conclusion rather than a separate section (Lines 601-611).

*Specific Comments*

– *Lines 41-43 "RANS only resolves the mean flow, and entirely relies on turbulence modelling to incorporate the effect of fluctuating components. As a result, RANS often fails to accurately reproduce transient flow features and quantities such as turbulent kinetic energy (van Hooff et al., 2017)". It is not evident that the RANS characteristics prevent them to
235     reproduce transient flow features. Do you mean unsteady flow features? Concerning turbulent kinetic energy predictions with RANS Models probab;y there are models that claim they can do this. See for example: Bartzis, J,G , Boundary-Layer Meteorology (2005) 116: 445–459 ,DOI 10.1007/s10546-004-7404-y*

By the phrase 'transient flow features', we indeed meant the 'unsteady flow features' *i.e.* the time dependent evolution of the flow. While simulating the cross-ventilation for an isolated building, van Hooff et al. (2017) showed that all RANS models, namely shear stress transport (SST), renormalization group (RNG) or Reynolds stress model (RSM), failed in reproducing the turbulent kinetic energy because steady RANS could not capture the vertical oscillation of the jet entering through the windward window of the building. We agree with the reviewer that the original statement was too broad. We have rephrased the sentence in the revised manuscript to "... these models often fail to reproduce unsteady flow features, leading to inaccurate prediction of quantities like turbulent kinetic energy for flows that involve time dependent modulation (Moonen et al., 2012; van Hooff et al., 2017; Blocken, 2018; Vita et al., 2020)" (Lines 40-42).

– *Lines 45-46: "... LES is inherently more accurate than RANS....". It would be better to say "... LES is expected to be more accurate than RANS". We should not forget that we have to apply LES correctly and this not always easy for open flows such as the atmospheric boundary layer.*

We have changed the sentence on Lines 47-49 to ".. [LES] is expected to be more accurate than RANS ... provided that the boundary conditions, including the surface, are modelled accurately, which is challenging for urban boundary layer flows."

– *Lines 105-107: Concerning horizontal grid, it seems that there is a different treatment for $x$- and $y$-directions. In real problems this might impose limited degree of freedom in selecting $x$- and $y$- directions. In other words is this selection application dependent?*

In practice, it is true that the $x$- and $y$- directions almost always conceptually correspond to the spanwise and streamwise directions respectively. However, this does not have to be the case when running periodic simulations, and when using inflow-outflow boundaries with fixed values. When using a time-varying inflow condition though, this can only be specified on the $x$ inlet plane. We have included this discussion in the Conclusion (Lines 607-609).

– *Lines 110-113: The inlet flow conditions are briefly discussed. The key problem here is how the upwind large eddies are introduced into the computation domain, keeping in mind that any introduced error persists downwind. Please expand.*

We have added (on Line 117-121): "The resolution in the $y$- and $z$-directions of the main simulation necessarily equals that of the precursor, meaning there no interpolation is required in these directions, and the flow variables on the output plane of the precursor can be copied directly to the inlet of the main simulation. Technically, the Dirichlet inflow boundary condition for the main simulation is enforced at $x = 0$, so $u$ can be set directly as it defined on cell edges. The other variables are (linearly) interpolated in the $x$-direction as they are defined on cell centres."

Alternatively, the reviewer may be referencing the need for a 'run-up' region before the flow reaches the geometry. We have also added on Lines 122-124: "It is good practice to have a reasonable distance between the boundaries and the object of interest, in order to allow the flow to adjust (Tominaga et al., 2008)."

**Other changes**

– Line 11: changed 'a previous version of uDALES' to 'a grid-aligned geometry'.

– Restructured the first paragraph of the Introduction in order to make the motivation clearer.

- Lines 38-49: expanded discussion of RANS modelling in the Introduction in order to contrast with LES.

- Lines 60-65: Revised discussion of Fig. 1.

- Line 124: defined symbols $L_x$, $L_y$, and $L_z$ as this was missing in the previous manuscript.

- Sect. 2.2: rephrased for clarity, and defined variables in a more specific manner.

- Table 1: changed caption to 'Comparison of upgraded features between uDALES v1.0 and v2.0.'

- Sect. 3.1: changed 'points' to 'cells' throughout for consistency, and simulation sizes specified more clearly (Lines 186-187 and 196-197). Also, added a short conclusion (Lines 200-201).

- Figure 4: moved from end of Sect. 3.1 to start.

- Lines 263-266: rephrased for clarity.

- Equation (9): changed subscript $m$ to $f$ to avoid ambiguity with usage in Eq. (8).

- Sect. 3.4: added more detail to the introduction (Lines 333-336), added subscript $f$ where appropriate, added a sentence on the accuracy of the new method for direct solar radiation (Lines 364-366).

- Sect. 3.5: changed 'driver' to 'precursor' throughout.

- Table 2: transposed the data for formatting purposes. Also, further detail added to caption.

- Line 489: added a more up-to-date and relevant reference.

- Figs. 8-13: added detail to captions.

- Added slightly more detail on future radiation modelling in the Conclusion (Lines 621-623).

- Line 627: changed Zenodo link and added reference.

- Minor wording changes throughout text for consistency and clarity.

**References**

Anders, J., Schubert, S., Sauter, T., Tunn, S., Schneider, C., and Salim, M.: Modelling the impact of an urban development project on microclimate and outdoor thermal comfort in a mid-latitude city, Energy and Buildings, 296, 113 324, 2023.

Bartholomew, P., Deskos, G., Frantz, R. A., Schuch, F. N., Lamballais, E., and Laizet, S.: Xcompact3D: An open-source framework for solving turbulence problems on a Cartesian mesh, SoftwareX, 12, 100 550, 2020.

Blocken, B.: LES over RANS in building simulation for outdoor and indoor applications: A foregone conclusion?, in: Building Simulation, vol. 11, pp. 821–870, Springer, 2018.

Grylls, T. and van Reeuwijk, M.: Tree model with drag, transpiration, shading and deposition: Identification of cooling regimes and large-eddy simulation, Agricultural and Forest Meteorology, 298, 108 288, 2021.

Hefny, M. M. and Ooka, R.: CFD analysis of pollutant dispersion around buildings: Effect of cell geometry, Building and Environment, 44, 1699–1706, 2009.

Krč, P., Resler, J., Sühring, M., Schubert, S., Salim, M. H., and Fuka, V.: Radiative Transfer Model 3.0 integrated into the PALM model system 6.0, Geoscientific Model Development, 14, 3095–3120, 2021.

Li, N. and Laizet, S.: 2decomp & fft-a highly scalable 2d decomposition library and fft interface, in: Cray user group 2010 conference, pp. 1–13, 2010.

Moonen, P., Dorer, V., and Carmeliet, J.: Effect of flow unsteadiness on the mean wind flow pattern in an idealized urban environment, Journal of wind engineering and industrial aerodynamics, 104, 389–396, 2012.

Rolfo, S., Flageul, C., Bartholomew, P., Spiga, F., and Laizet, S.: The 2DECOMP&FFT library: an update with new CPU/GPU capabilities, Journal of Open Source Software, 8, 5813, 2023.

Salim, M. H., Schlünzen, K. H., Grawe, D., Boettcher, M., Gierisch, A. M., and Fock, B. H.: The microscale obstacle-resolving meteorological model MITRAS v2. 0: model theory, Geoscientific Model Development, 11, 3427–3445, 2018.

Salim, M. H., Schubert, S., Resler, J., Krč, P., Maronga, B., Kanani-Sühring, F., Sühring, M., and Schneider, C.: Importance of radiative transfer processes in urban climate models: a study based on the PALM 6.0 model system, Geoscientific Model Development, 15, 145–171, 2022.

Suter, I., Grylls, T., Sützl, B. S., Owens, S. O., Wilson, C. E., and van Reeuwijk, M.: uDALES 1.0: a large-eddy simulation model for urban environments, Geoscientific Model Development, 15, 5309–5335, 2022.

Tominaga, Y., Mochida, A., Yoshie, R., Kataoka, H., Nozu, T., Yoshikawa, M., and Shirasawa, T.: AIJ guidelines for practical applications of CFD to pedestrian wind environment around buildings, Journal of wind engineering and industrial aerodynamics, 96, 1749–1761, 2008.

van Hooff, T., Blocken, B., and Tominaga, Y.: On the accuracy of CFD simulations of cross-ventilation flows for a generic isolated building: Comparison of RANS, LES and experiments, Building and Environment, 114, 148–165, 2017.

Vita, G., Salvadori, S., Misul, D. A., and Hemida, H.: Effects of inflow condition on RANS and LES predictions of the flow around a high-rise building, Fluids, 5, 233, 2020.

Xie, Z.-T. and Castro, I. P.: Efficient generation of inflow conditions for large eddy simulation of street-scale flows, Flow, turbulence and combustion, 81, 449–470, 2008.